# Ultrasound-responsive low-dose doxorubicin liposomes trigger mitochondrial DNA release and activate cGAS-STING-mediated anti-tumour immunity

Chaoyu Wang [1,2], Ruoshi Zhang[1,2], Jia He[1,2], Lvshan Yu[1,2], Xinyan Li[1,2], Junxia Zhang[2,3,4], Sai Li [2,3,4], Conggang Zhang [1,2], Jonathan C. Kagan [5], Jeffrey M. Karp [6,7,8,9] & Rui Kuai [1,2] ✉

DNA derived from chemotherapeutics-killed tumor cells is one of the most important damage-associated molecular patterns that can activate the cGAS-STING (cyclic GMP-AMP synthase−stimulator of interferon genes) pathway in antigen-presenting cells (APCs) and promote antitumor immunity. However, conventional chemotherapy displays limited tumor cell killing and ineffective transfer of stable tumor DNA to APCs. Here we show that liposomes loaded with an optimized ratio of indocyanine green and doxorubicin, denoted as LID, efficiently generate reactive oxygen species upon exposure to ultrasound. LID plus ultrasound enhance the nuclear delivery of doxorubicin, induce tumor mitochondrial DNA oxidation, and promote oxidized tumor mitochondrial DNA transfer to APCs for effective activation of cGAS-STING signaling. Depleting tumor mitochondrial DNA or knocking out STING in APCs compromises the activation of APCs. Furthermore, systemic injection of LID plus ultrasound over the tumor lead to targeted cytotoxicity and STING activation, eliciting potent antitumor T cell immunity, which upon the combination with immune check-point blockade leads to regression of bilateral MC38, CT26, and orthotopic 4T1 tumors in female mice. Our study sheds light on the importance of oxidized tumor mitochondrial DNA in STING-mediated antitumor immunity and may inspire the development of more effective strategies for cancer immunotherapy.

Chemotherapeutics such as doxorubicin can induce immunogenic cell death (ICD) of tumor cells[1–3]. The key features of chemotherapeutics-induced ICD include CRT exposure on dying tumor cells and release of damage-associated molecular patterns (DAMPs) such as HMGB1, ATP, and the most recently identified DAMP tumor DNA[4]. When antigen-presenting cells (APCs) acquire tumor antigens and damage-associated molecular patterns (DAMPs) from tumor cells undergoing ICD, they can present tumor antigen epitopes, upregulate costimulatory signals,

[1]School of Pharmaceutical Sciences, Tsinghua University, Beijing 100084, China. [2]Tsinghua-Peking Center for Life Sciences, Beijing 100084, China. [3]School of Life Sciences, Tsinghua University, Beijing 100084, China. [4]Frontier Research Center for Biological Structure & State Key Laboratory of Membrane Biology, Beijing 100084, China. [5]Division of Gastroenterology, Boston Children's Hospital and Harvard Medical School, Boston, MA, USA. [6]Department of Anesthesiology, Perioperative, and Pain Medicine, Brigham and Women's Hospital, Harvard Medical School, Boston, MA, USA. [7]Harvard-MIT Program in Health Sciences and Technology, MIT, Cambridge, MA, USA. [8]Harvard Stem Cell Institute, Harvard University, Cambridge, MA, USA. [9]Broad Institute of MIT and Harvard, Cambridge, MA, USA. ✉e-mail: ruikuai@tsinghua.edu.cn

and secrete cytokines for activation of antitumor T cells, especially cytotoxic CD8 + T cells[5]. During this complex process, DAMPs from tumor cells play critical roles in shaping the antitumor T cell immunity[6,7]. Recent studies have shown that dying tumor cell-derived DNA as a DAMP can activate the cytosolic DNA sensor cyclic GMP-AMP synthase (cGAS) in APCs to convert ATP and GTP to cGAMP[8,9], which functions as a second messenger to activate the endoplasmic reticulum (ER) adaptor STING[4,10,11] to induce secretion of type I interferons and enhance the activation of antitumor T cell immunity[4,12–14]. The use of chemotherapeutics to kill tumor cells and utilize DNA from dying tumor cells for activating T cell immunity holds great potential for highly effective cancer immunotherapy. However, conventional chemotherapeutics have limited tumor-targeted drug delivery, with only ~1% injected dose reaching the tumor following systemic injection[15,16]. Consequently, traditional chemotherapies have poor tumor cell killing in vivo, which can prevent the transport of tumor DNA to APCs in situ for effective STING activation. Furthermore, the presence of DNase in APCs can degrade tumor DNA, which can completely lose its activity after degradation[17]. Therefore, it is urgent to improve the tumor-killing effect of chemotherapeutics in a safe and effective manner and stabilize DNA in order to unleash the immunostimulatory potential of tumor DNA as a DAMP for cancer immunotherapy.

Intratumoral or peritumoral injections of chemotherapeutics-loaded nanoparticles or hydrogels have been used to successfully enhance the ICD of tumor cells in vivo, but these approaches are typically used for surface tumors[18,19]. Systemic injection of chemotherapeutics-loaded nanoparticles is broadly applicable to tumors at different locations, and introducing targeting ligands to these nanoparticles has shown enhanced tumor targeting[20,21]. However, the endosomal/lysosomal sequestering of nanoparticles can prevent the chemotherapeutics reaching the target[22]. Furthermore, these approaches did not directly address the tumor DNA degradation issue.

Inspired by the fact that phagocytes generate reactive oxygen species (ROS) to kill pathogens and introduce oxidative modifications such as 8-hydroxy-2′-deoxy-guanosine (8-OHdG) to DNA to make it more resistant to DNase III[23,24], we sought to take advantage of ROS to enhance the tumor-killing effect of chemotherapeutics and oxidize tumor DNA to improve its stability. Recently, sonodynamic therapy (SDT), which is typically achieved by exposing sonosensitizers to safe and non-radioactive ultrasound (US) to generate ROS in the desired location to induce cancer cell apoptosis and/or necrosis, is emerging as an attractive strategy for cancer therapy[25–27]. While SDT itself has shown some potential for tumor cell killing and immune activation in vivo, the overall efficiency remains very limited. Combining SDT with exogenous immunostimulatory agents has been shown to improve the immune activation and therapeutic efficacy[25], but how to fully unleash the immunostimulatory potential of endogenous tumor DNA itself remains unexplored. Compared with exogenous immunostimulatory agents, efficient use of endogenous DAMPs, especially tumor DNA will allow for more efficient immune activation in the tumor region without causing systemic side effects. Due to the good safety, non-radioactivity, and deep penetration of ultrasound, the SDT-induced ROS can be a useful tool to tune the activity of chemotherapeutics, enhance tumor DNA stability, and promote DNA transfer to APCs in a highly controllable manner. To achieve these goals, we aimed to load sonosensitizers and chemotherapeutics in nanoscale liposomes, which have a track record of good safety and stability. We chose the FDA approved Indocyanine green (ICG) as the sonosensitizer[28] and doxorubicin (DOX) as the chemotherapeutic for loading in liposomes to obtain liposomal ICG/DOX, denoted as LID, with an ultimate goal of unleashing the potential of tumor DNA for safe and effective STING activation and cancer immunotherapy (Fig. 1).

Here, we show that upon exposure to US, LID efficiently generates ROS, which enhances the delivery of DOX to nuclei of tumor cells, and

functionally synergizes with DOX to kill tumor cells. LID + US efficiently oxidizes tumor mitochondrial DNA and facilitates the transfer of the oxidized tumor mitochondrial DNA to APCs, inducing STING activation in APCs. Systemic injection of LID followed by US treatment over the tumor region exhibits targeted tumor cell killing and STING activation in the tumor, resulting in potent antitumor T cell immunity. Notably, the combination of LID + US with αPD-L1, which has been widely used to reverse the immune suppression on T cells[29–31], regresses multiple types of tumors in female mice. Our study has also advanced the understanding of oxidized tumor mitochondrial DNA in STING-mediated antitumor immunity and may inspire the development of more effective strategies for the treatment of cancer.

## Results
### Preparation and characterization of LID
We first developed an easily scalable protocol for the preparation of liposomal ICG/DOX (LID) based on the active drug loading mechanism for DOX[32,33]. Briefly, ethanol containing lipids was mixed with 250 mM $(NH_4)_2SO_4$ and extruded through 100-nm polycarbonate membranes to obtain blank liposomes, which were then passed through a PD-10 column to remove external $(NH_4)_2SO_4$ and incubated with doxorubicin (DOX) at 55 °C for 30 min to allow for the $(NH_4)_2SO_4$ gradient-driven active loading of DOX into liposomes. To load ICG into liposomes, ICG was first conjugated to DOPE to obtain DOPE-ICG (Supplementary Fig. 1) before incubation with liposomes at room temperature for 30 min so that DOPE could anchor ICG onto the surface of liposomes (Fig. 1a). LID had over 90% loading efficiency of DOX and over 95% loading efficiency of ICG, respectively (Fig. 1b), with an average diameter around 130 nm (Fig. 1c). Cryo-electron microscopy (cryo-EM) confirmed the homogeneous size distribution of LID and the presence of low-dose DOX nanocrystals in the aqueous core of LID (Fig. 1d). Liposomal ICG (LI) and Liposomal DOX (LD) exhibited similar sizes, and LD, but not LI, had DOX nanocrystals in the aqueous core (Supplementary Fig. 2 and Supplementary Movie 1). LID exhibited minimal DOX release and ICG release in PBS or FBS-containing buffer at 37 °C over 48 h (Supplementary Fig. 3), indicating the good stability of LID.

### LID + US induced efficient ROS generation and intracellular DOX delivery
To evaluate ROS generation of LID, we incubated LID or other control formulations with CT26 tumor cells for 24 h and then replaced the old medium with fresh medium containing ROS sensing probe DCFH-DA, which became fluorescent after oxidation by ROS. LID + US exhibited potent ROS generation, and the ROS generation was positively correlated with ultrasound time over the range of 1–5 min (Fig. 2a). In particular, LID and 5 min of ultrasound induced 14.2-fold more ROS than LI and 15.2-fold more ROS than LD (P < 0.0001). LID + US exhibited similar levels of ROS generation compared with LI + US when the ultrasound time was identical, indicating DOX did not interfere with the ROS generation. Ultrasound alone also induced some ROS, but the level was 2.2-fold and 1.95-fold lower than LID + US and LI + US (P < 0.0001, Fig. 2a), respectively, indicating the sonosensitizer ICG was critical for generating ROS. Confocal microscopy showed that LI + US and LID + US both induced elevated intracellular ROS, which was evenly distributed inside CT26 tumor cells, while LD itself did not significantly increase ROS (Fig. 2b). We found similar ROS generation patterns in MC38 tumor cells treated with indicated formulations (Supplementary Fig. 4). Notably, following the uptake of liposome formulations, ICG was mostly located in endosomes (Supplementary Fig. 5), where liposomes are typically trapped. The differential distribution profile of ICG and ROS indicates that ROS can escape from endosomes and reach other parts of cells.

To track the intracellular delivery of DOX, we incubated LID or other control formulations with MC38 cells for 24 h and then applied ultrasound to indicated groups. After another 24 h, we imaged the

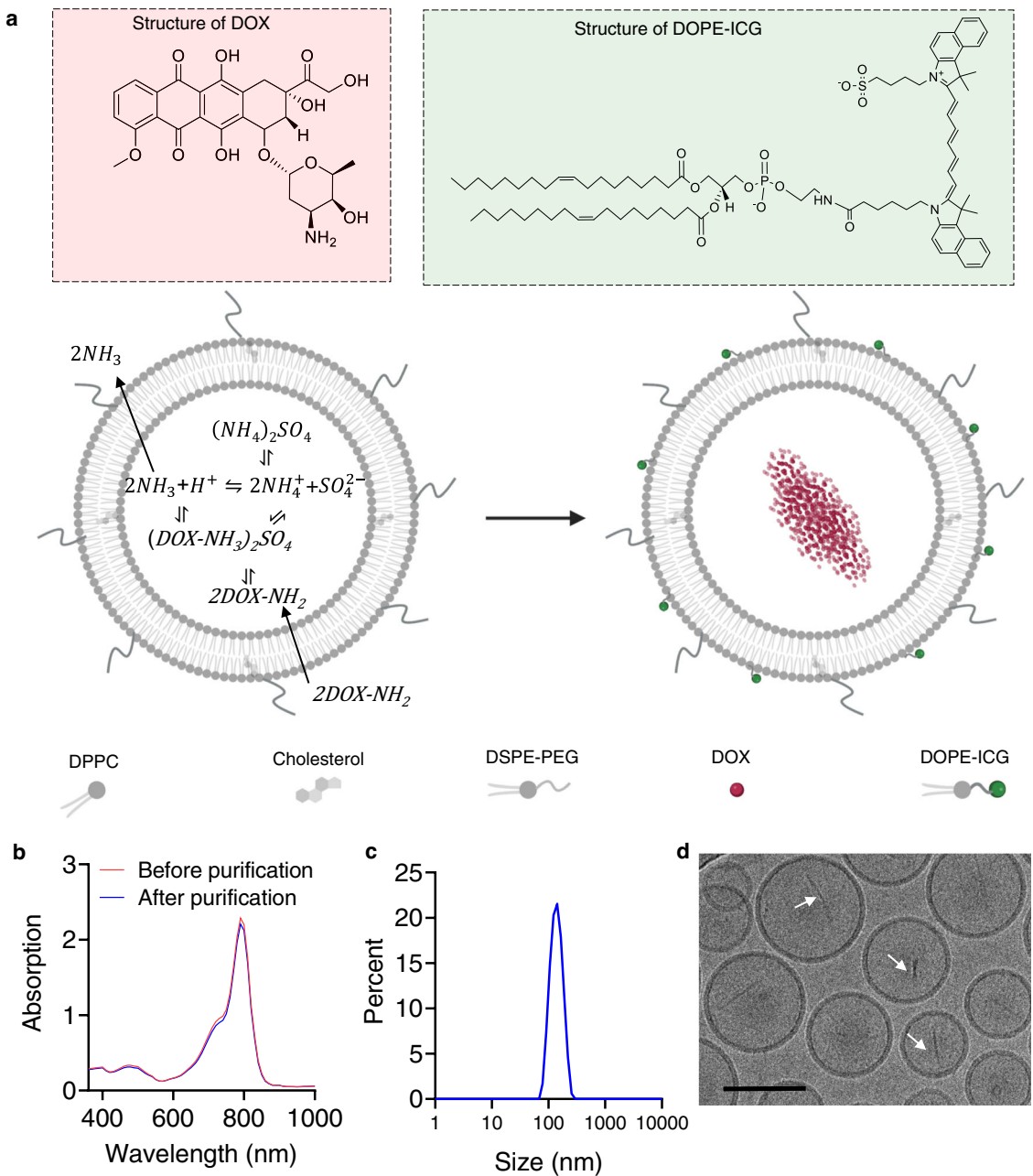

**Fig. 1 | Preparation and characterization of liposomal ICG/DOX (LID).**
**a** Schematic showing the preparation of LID. Blank liposomes were prepared with 250 mM $(NH_4)_2SO_4$, followed by removal of external $(NH_4)_2SO_4$ using size exclusion chromatography to establish the transmembrane gradient. DOX was incubated with the blank liposomes at 55°C to enable drug loading. When $NH_3$ escaped liposomes, one $H^+$ was produced and retained in the liposome, resulting in an acidic core. When DOX diffused into the liposome, it became protonated and trapped within the liposome. As DOX loading into the liposome transiently increased the internal pH, it further increased the level of ammonia and created more $H^+$, allowing more DOX to be loaded into the liposome. Ultimately, DOX forms a crystalline precipitate due to the presence of sulfate anions inside the liposome. ICG was covalently attached to DOPE before incubation with DOX-loaded liposomes such that DOPE could anchor ICG onto liposomes. **b** The absorption spectrum of LID. **c** representative size distribution of LID measured by dynamic laser scattering (DLS). **d** Cryo-electron microscopy (cryo-EM) of LID, scale bar = 100 nm. White arrows indicate low-dose DOX nanocrystals. The data are representative of two independent experiments (**b**–**d**). Source data are provided as a Source Data file.

fluorescence signal of DOX inside MC38 cells by confocal microscopy. The control group or LI + US did not show any DOX signal due to the absence of DOX in the formulation. Cells treated with LD had the most DOX signal around the perinuclear region. This pattern was consistent with previous studies showing endosomal/lysosomal sequestering of liposomal doxorubicin following cellular uptake[22]. Strikingly, cells treated with LID + US had the most DOX signal in the nuclei (Fig. 2c). It should be noted that both LD and LID + US had similar levels of total DOX inside tumor cells, indicating that LID + US did not change the cellular uptake of DOX, but instead redistributed DOX to the nuclei, where the target of DOX is located. Further analysis indicated that under neutral pH, LID released minimal DOX in the absence or presence of ultrasound. Under acidic pH (endosomal pH), LID efficiently released DOX and US only slightly boosted DOX release (Supplementary Fig. 6a). As ICG was covalently attached to the phospholipid, LID released minimal ICG under neutral pH or acidic pH. US boosted ICG release at acidic pH much more than that of neutral pH, but the overall release was much lower than DOX (Supplementary Fig. 6b). These

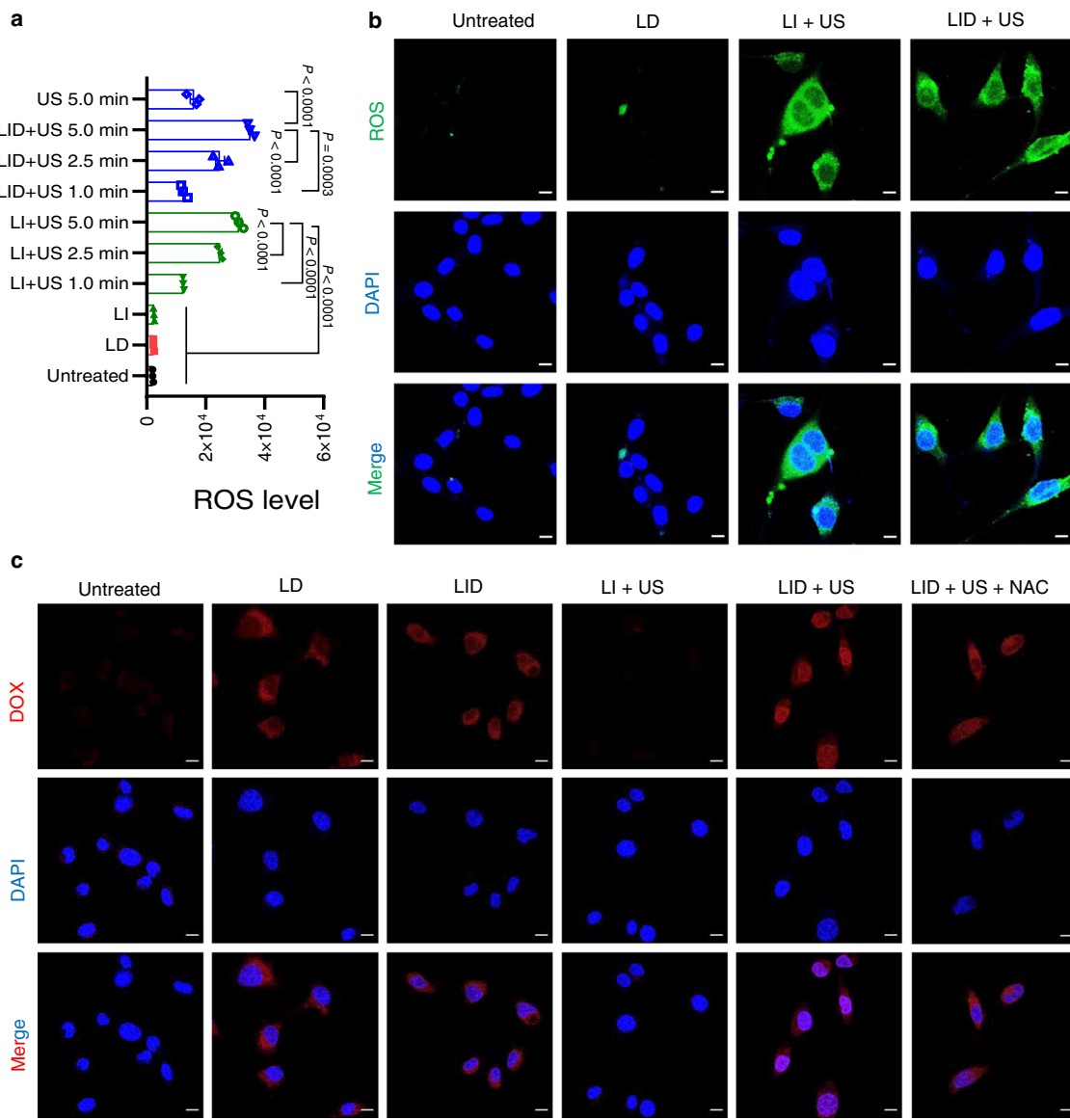

**Fig. 2 | LID + US efficiently generated ROS and promoted DOX delivery to the nuclei of tumor cells. a** ROS generation in CT26 cells induced by indicated formulations with or without ultrasound (US). The data were analyzed by one-way analysis of variance (ANOVA) with Tukey's multiple comparisons post test. Data represent mean ± SEM (*n* = 3 experimental replicates per group). **b** Confocal images showing ROS generation in CT26 tumor cells induced by indicated formulations with or without ultrasound (US). Scale bars, 10 µm. **c** MC38 tumor cells were incubated with indicated formulations for 24 h, and then ultrasound (US) was applied to selected groups. After another 24 h, intracellular delivery of DOX was imaged by confocal microscopy. Scale bars, 10 µm. The data are representative of two independent experiments (**b**, **c**). Source data are provided as a Source Data file.

results imply that the slightly accelerated DOX release triggered by LI + US was not the major reason for the nuclear distribution of DOX. In fact, as an organic base, released DOX was protonated within acidic endosomes and unable to efficiently cross the endosomal membrane[33]. Since ICG was available in endosomes to generate ROS in response to US and ROS can increase permeability of endosomes[27], we sought to further analyze the permeability of endosomal membrane. We took advantage of a fluorescent dye Acridine Orange (AO), which was able to cross the plasma membrane of live cells and exhibit red fluorescence within acidic endosomes but green fluorescence within neutral nuclei. Importantly, AO was protonated within acidic endosomes and unable to cross the endosomal membrane unless the permeability of endosomes was increased, so the disappearance of red AO fluorescence has been used to evaluate the endosomal membrane permeability[34]. We treated MC38 cells with LI or LI + US and then added AO to MC38 cells. We found AO + LI led to bright red fluorescence in endosomes, while AO + LI + US did not exhibit red fluorescence in

endosomes (Supplementary Fig. 7), indicating LI + US enhanced the permeability of the endosomal membrane, which facilitated the endosomal escape of DOX and its delivery to the nuclei. Indeed, LID + US in combination with the ROS scavenger NAC reduced DOX signal in the nuclei, further indicating ROS was critical in promoting the nuclear delivery of DOX. In line with the enhanced DOX delivery to the nuclei, LID + US also induced more DNA breaks in the nucleus such as H2AX foci compared with LI + US and LD (Supplementary Fig. 8).

## LID + US efficiently killed tumor cells in vitro
To test the tumor-killing effect of LID in vitro, we treated CT26 tumor cells with LID or other control formulations for 24 h and then applied ultrasound to indicated groups. In the presence of 1 min of ultrasound, all LID + US groups containing different ratios of ICG/DOX had significantly lower viabilities compared with the untreated group (*P* < 0.0001), and increasing the weight ratio of ICG/DOX in LID from 1:1 to 8:1 enhanced the tumor cell killing effect (*P* < 0.0001, Fig. 3a). A ratio

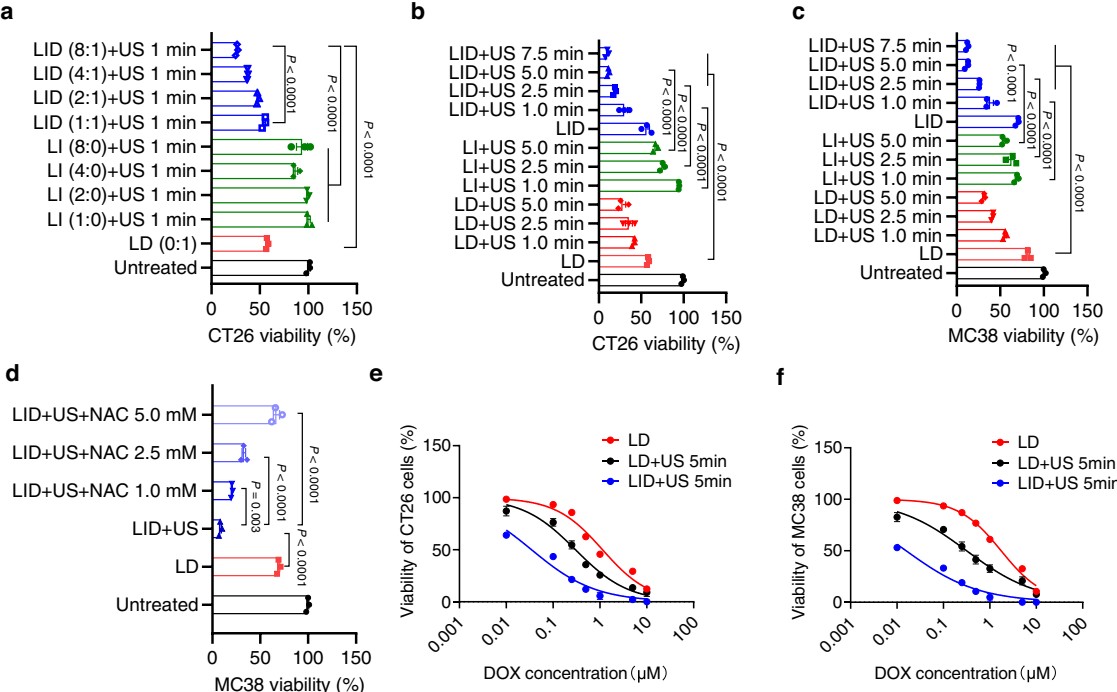

**Fig. 3 | Ultrasound responsive chemotherapeutics exhibited potent tumor cell killing in vitro in an ultrasound dependent manner. a** CT26 tumor cells were treated with indicated formulations containing different concentrations of ICG/DOX for 24 h, and then ultrasound (1 min) was applied to selected groups. After 24 h, the cell viability was measured by the cell counting kit. Numbers in the parenthesis indicate the weight ratio of ICG and DOX ($n = 3$ experimental replicates per group). **b, c** Viability of CT26 or MC38 tumor cells treated with indicated formulations with or without ultrasound. ICG was fixed at 3 µM, DOX was fixed at 0.5 µM, and ultrasound exposure time was between 1 min and 7.5 min ($n = 3$ experimental replicates per group). **d** Effect of ROS scavenger NAC on the viability of MC38 cells treated with indicated formulations ($n = 3$ experimental replicates per group). **e, f** IC50 of DOX in CT26 or MC38 tumor cells for indicated formulations ($n = 3$ experimental replicates per group). The data represent mean ± SEM. **a–f** Data were analyzed by one-way analysis of variance (ANOVA) with Tukey's multiple comparisons post test. Source data are provided as a Source Data file.

of ICG/DOX higher than 8:1 was not tested as it is beyond the loading capacity of liposomes. In contrast, under this condition, LI + US failed to further reduce the cell viability compared with the untreated group, indicating the presence of DOX was critical. When the weight ratio of ICG/DOX was 8:1 in LID, LID + US exhibited better tumor cell killing than LD ($P < 0.0001$, Fig. 3a). Based on these results, we used the 8:1 weight ratio of ICG/DOX for subsequent experiments. We next investigated the impact of ultrasound time on tumor cell killing. The tumor cell killing effect of LID + US was positively correlated with ultrasound time over the time range of 1–5 min, and LID + US exhibited better CT26 tumor cell killing than LD ($P < 0.001$) and LI + US ($P < 0.0001$; Fig. 3b). Increasing the ultrasound time to 7.5 min did not further enhance the tumor cell killing, so the ultrasound time of 5 min was used for subsequent experiments. Moreover, LID alone or LD + US exhibited a weaker tumor-killing effect compared with LID + US, indicating ultrasound and ICG are required for efficient killing of tumor cells (Fig. 3b). We found a similar pattern when treating MC38 tumor cells with these formulations (Fig. 3c). Notably, depleting ROS with NAC significantly compromised the efficacy of LID + US, indicating that ROS was crucial in mediating the tumor-killing effect of LID + US (Fig. 3d). We next broadened the concentration range of DOX and evaluated their cytotoxicity in the form of LD, LD + US, or LID + US. Strikingly, for CT26 tumor cells, LID had an IC50 of 0.035 µM DOX, which was 32.7-fold lower than the IC50 of LD, a formulation similar to the commercially available Doxil ®, and 8.7-fold lower than LD + US (Fig. 3e). Similarly, for MC38 tumor cells, LID had an IC50 of 0.015 µM DOX, which was 111.3-fold lower than the IC50 of LD and 20.1-fold lower than that of LD + US (Fig. 3f). Moreover, LID + US had much lower IC50 on other tumor cells such as HCT116, Hela, and MDA-MB-231 cells compared with LD and LD + US (Supplementary Fig. 9). These results indicate that LID + US can potently kill tumor cells in vitro and may be broadly applicable to different tumors.

## LID + US induced efficient tumor DNA oxidation and transport to DC

As oxidized DNA exhibits better stability in the presence of DNase[24], we sought to enhance the tumor DNA stability by inducing DNA oxidation. We treated CT26 tumor cells with LID or other control formulations for 24 h and then applied ultrasound to indicated groups. We then used a fluorescent antibody to label 8-OHdG, a hallmark of DNA oxidation. Confocal microscopy showed that both LI + US and LID + US induced DNA oxidation, while the untreated control and LD did not show a detectable level of DNA oxidation. Interestingly, most oxidized DNA was not in the nuclei, but in mitochondria (Fig. 4a). We speculate this is because the presence of additional barriers such as histone in the nuclei may protect nuclear DNA from oxidation[35]. To evaluate the tumor DNA transport to DC, we first treated MC38 tumor cells as described above and then added BMDCs. After co-culture for 24 h, we used a fluorescent antibody to label the 8OHdG of oxidized DNA and CD11c on DC. Confocal microscopy revealed that only LID + US induced efficient oxidized DNA transport to DC, while LD or LI + US failed to achieve this (Fig. 4b). We found a similar pattern for the transport of CT26 tumor DNA to DC (Supplementary Fig. 10). As both LI + US and LID + US induced tumor DNA oxidation, the difference in tumor DNA transport to DC implied that efficient tumor killing was critical for enhancing DNA transport from tumor cells to APCs.

## LID + US efficiently induced STING activation and antigen presentation

To investigate STING activation, we first treated MC38 tumor cells with LID or other control formulations for 24 h and then applied ultrasound to indicated groups. After another 24 h, we added RAW-Lucia™ ISG cells, which express many pattern recognition receptors (PRRs) and can sense cytosolic DNA to express luciferase. The ISRE reporter

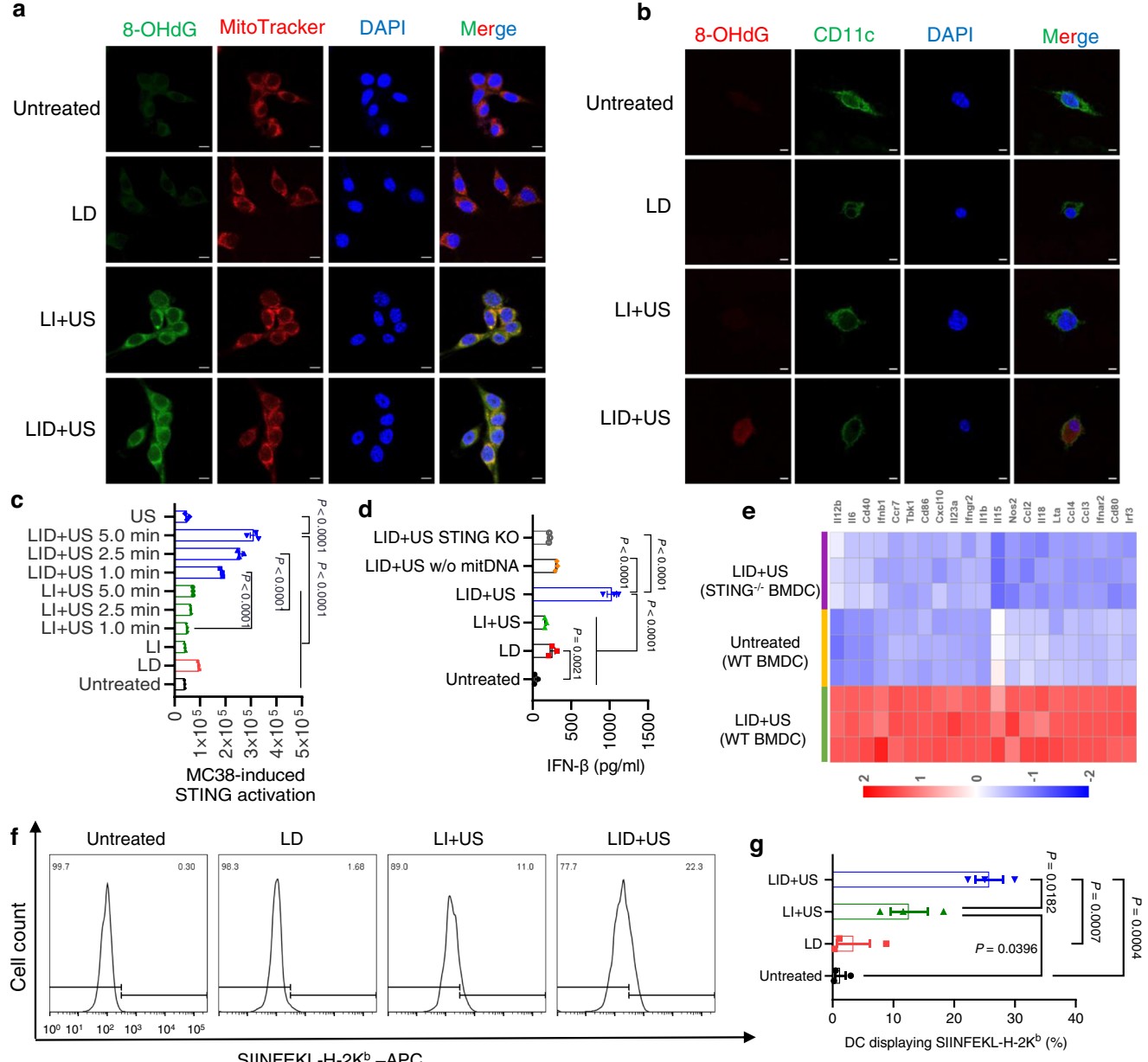

**Fig. 4 | Tumor cells killed by ultrasound-responsive chemotherapeutics triggered STING activation and antigen presentation. a** CT26 tumor cells were treated with indicated formulations for 24 h. Ultrasound (2 W/cm², 50% duty cycle, 5 min) was applied to selected groups. After 24 h, the cells were stained by an anti-8OHdG antibody and MitoTracker to label oxidized tumor DNA and mitochondria before confocal microscopy. Scale bars, 10 μm. **b** MC38 tumor cells were treated with indicated formulations for 24 h. Ultrasound (2 W/cm², 50% duty cycle, 5 min) was applied to selected groups. After 24 h, BMDCs were added and co-cultured for another 24 h, followed by staining with anti-8OHdG and CD11c antibodies before confocal microscopy. Scale bars, 5 μm. The data are representative of two independent experiments (**a**, **b**). **c** MC38-OVA tumor cells were treated with indicated formulations for 24 h. Different lengths of ultrasound (2 W/cm², 50% duty cycle) were applied to selected groups. After 24 h, RAW-lucia™ ISG reporter cells were added and co-cultured for another 24 h, followed by measuring the luminescence signal from RAW-lucia™ ISG reporter cells (*n* = 3 experimental replicates per group). **d** MC38 tumor cells with mitochondria DNA depleted by using

dideoxycytidine (ddC) were treated with indicated formulations for 24 h. Ultrasound (2 W/cm², 50% duty cycle, 5 min) was applied to selected groups. After 24 h, BMDCs were added and co-cultured for another 24 h, followed by measuring IFNβ using the ELISA kit (*n* = 3 experimental replicates per group). **e** MC38 tumor cells were treated with indicated formulations. After 24 h, WT BMDC or STING⁻/⁻ BMDC were added to tumor cells and co-cultured for another 24 h, followed by RNA-seq of BMDC (*n* = 3 experimental replicates per group). **f, g** MC38-OVA cells were treated with indicated formulations for 24 h. Then ultrasound (2 W/cm², 50% duty cycle, 5 min) was applied to selected groups. After 24 h, BMDCs were added and co-cultured for 24 h before antigen presentation on BMDCs was measured by flow cytometry. Shown are (**f**) representative histograms of antigen presentation on BMDCs and (**g**) percentages of BMDCs presenting the antigen epitopes (*n* = 3 experimental replicates per group). The data represent mean ± SEM (**c**, **d**, **g**). Data were analyzed by one-way analysis of variance (ANOVA) with Tukey's multiple comparisons post test. Source data are provided as a Source Data file.

activity of RAW-Lucia™ ISG cells induced by LID + US was positively correlated with ultrasound time over the time range of 1–5 min. In particular, when the ultrasound time was 5 min, LID + US exhibited 3.3-fold better activation than LD (P < 0.0001) and 4.3-fold better

activation than LI + US (P < 0.0001). LI or US alone did not show a detectable level of activation (Fig. 4c). These results were in line with the enhanced tumor cell killing and enhanced transport of oxidized tumor DNA to DC induced by LID + US. To further validate STING

activation in primary antigen-presenting cells, we first treated MC38 tumor cells as described above and then added BMDCs to dying tumor cells, a procedure mimicking fresh APCs are recruited to the damaged tumor tissue where they can acquire DNA from dying tumor cells[25]. After 24 h, we detected IFNβ in the supernatant as a readout for STING activation. BMDCs treated with LID + US secreted 4-fold and 6.25-fold more IFNβ than those treated with LD and LI + US, respectively ($P < 0.0001$, Fig. 4d). Notably, either depleting mitochondrial DNA from tumor cells (Supplementary Fig. 11) or knocking out STING from BMDCs resulted in 3.32-fold and 4.76-fold reduction of IFNβ secretion for the LID + US group (Fig. 4d). These results indicate that mitochondrial DNA of tumor cells and STING in BMDCs were both critical in mediating IFNβ secretion. Western blot analysis showed that LID + US induced strong STING signaling activation in BMDC, and the activation was dependent on mitochondrial DNA and STING in BMDC (Supplementary Fig. 12). We found IFNβ secretion was mainly from BMDCs rather than from tumor cells, as removing BMDCs from the co-cultured cells almost abrogated IFNβ secretion (Supplementary Fig. 13a). RNA sequencing further confirmed that LID + US activated STING-related genes in BMDCs, and knocking out STING in BMDCs largely abrogated the effect (Fig. 4e). We next sought to understand whether tumoral cGAMP or DNA-mediated APC activation. For the LID + US group, knocking out cGAS from MC38 tumor cells did not compromise the IFNβ secretion from BMDC, indicating cGAS in MC38 was not required to mediate the APC activation. In contrast, knocking out cGAS from BMDC abrogated IFNβ secretion from BMDC (Supplementary Fig. 13b). As tumoral cGAMP can directly induce IFNβ secretion from BMDC without requiring functional cGAS in BMDC, these results indicate that tumor DNA, rather than tumoral cGAMP, mainly mediated the activation effect. Moreover, we also found that LID + US upregulated other ICD markers such as CRT and HMGB1, with levels similar (for CRT) or higher (for HMGB1) compared with LD and LI + US (Supplementary Fig. 14). While our data suggest that oxidized tumor mitochondrial DNA is important in STING-mediated antitumor immunity after treatment by LID + US, we can not rule out that other DAMPs (e.g., HMGB1 and genomic DNA) may also play a role. The effect of different DAMPs is highly dependent on the way how tumor cells are treated. To evaluate the antigen presentation on dendritic cells, we treated the model antigen protein ovalbumin expressing MC38 (MC38-OVA) tumor cells with LID or other control formulations for 24 h and then applied ultrasound to indicated groups. After 24 h, we added BMDCs to co-culture with the dying tumor cells for 24 h and then measured antigen presentation on BMDCs. LID + US induced 2-fold and 7.5-fold stronger antigen presentation compared with LI + US and LD, respectively, indicating LID + US also facilitated antigen presentation (Fig. 4f, g).

## LID + US induced a potent therapeutic effect and antitumor T cell immunity

The promising in vitro activity of LID + US motivated us to explore their therapeutic effect in vivo. We first examined the pharmacokinetics and biodistribution profile based on the intrinsic fluorescence signal from ICG. LID and LI exhibited similar long-circulating properties, while the free ICG was quickly cleared in vivo within 5 min (Fig. 5a). In line with the pharmacokinetics, LID and LI had more chance to accumulate in the tumor region through the enhanced permeability and retention (EPR) effect[15], while free ICG had marginal accumulation in the tumor. In particular, LID and LI had stable accumulation between 12 and 24 h after intravenous injection. After 24 h, the accumulation started to decrease (Fig. 5b). Based on the pharmacokinetics and biodistribution profile, we chose 24 h post intravenous injection as the time point for ultrasound treatment to minimize the side effect on blood cells while maintaining the therapeutic effect over the tumor region. Although LID also accumulated in other normal organs (Supplementary Fig. 15), LID in normal organs was not activated due to the absence of ultrasound.

For the treatment of subcutaneous MC38 tumors in female mice, we intravenously injected LID or other control formulations and applied ultrasound to the tumor region 24 h post injection of formulations (Fig. 5c). After two cycles of treatment, LID + US exhibited more potent tumor growth inhibition than LD and LI + US, both of which only modestly inhibited tumor growth (Fig. 5d, e). LI or LD + US had modest tumor growth inhibition, indicating the presence of sonosensitizer in LID + US was critical in mediating the therapeutic effect (Supplementary Fig. 16). Importantly, animals treated with all formulations did not cause body weight decrease or side effects in normal organs (Supplementary Fig. 17a, b). We used a similar dosing regimen to treat subcutaneous CT26 tumors. After three cycles of treatment, LID + US strongly inhibited the tumor growth, while LD and LI + US only showed some marginal therapeutic effect (Supplementary Fig. 18). To investigate the immune response, we established subcutaneous MC38-OVA tumors in female mice in a separate study and treated the animals as described above. Two days after the ultrasound treatment, LID + US induced 7.04-fold and 6.83-fold more DCs displaying SIINFEKL/H-2K$^b$ in the tumor compared with LD ($P < 0.01$) and LI + US ($P < 0.01$, Fig. 5f), respectively. LID + US also significantly increased the percentage of CD80 + DC in the tumor (Fig. 5g), increased the percentage of CD86 + DC in the tumor-draining lymph nodes (Fig. 5h), and enhanced the phosphorylation of TBK1 and IRF3, which were related to the activation of STING pathway in the tumor tissue (Fig. 5i). As LD and LI + US failed to significantly induce STING activation in the tumor, we focused on LID + US in terms of ISG levels. Compared with the untreated group, LID + US upregulated ISG levels (Supplementary Fig. 19). Accordingly, LID + US induced 1.94-fold more and 4.35-fold more CD8 + T cells in the tumor than LD and LI + US, respectively (Fig. 5j, k and Supplementary Fig. 20). Moreover, LD and LI + US exhibited similar levels of SIINFEKL-specific CD8 + T cells compared with the untreated animals, while LID + US exhibited ~30% SIINFEKL-specific CD8 + T cells (Fig. 5l). To decouple the STING pathway-mediated therapeutic effect from the direct tumor-killing effect of LID + US, we examined the therapeutic effect of LID + US on STING knockout (STING KO) mice bearing subcutaneous MC38 tumors. Knocking out STING in the host significantly compromised but not completely abrogated the therapeutic effect of LID + US (Fig. 5m). Interestingly, knocking out STING in MC38 tumor cells (Supplementary Fig. 21) but not C57BL/6 mice maintained the therapeutic effect of LID + US (Fig. 5n), indicating the STING pathway in the host rather than in tumor cells was mainly responsible for the therapeutic effect of LID + US. We performed additional experiments to test LID + US combined with CD8 + T cell depletion. Depleting CD8 + T cells significantly compromised the therapeutic effect of LID + US (Supplementary Fig. 22). Together with the compromised therapeutic effect of LID + US on STING KO mice, these data indicate that the host immune activation is important to mediate the antitumor effect in vivo.

## LID + US in combination with αPD-L1 eliminated bilateral tumors

Having seen the potent therapeutic effect of LID + US on subcutaneous MC38 tumors, we sought to evaluate its therapeutic efficacy on bilateral MC38 tumors in female mice[29,36], which have been widely used to mimic tumors that have metastasized to multiple sites. We intravenously injected LID to tumor-bearing mice and applied ultrasound to the primary tumor 24 h post injection of formulations (the distant tumor was not exposed to ultrasound). We also intraperitoneally injected the αPD-L1 antibody to indicated groups (Fig. 6a). αPD-L1 alone showed a marginal therapeutic effect on the primary tumor compared with untreated animals. LID + US strongly inhibited the tumor growth, but did not completely regress the primary tumor. Remarkably, LID + US + αPD-L1 resulted in complete regression of the primary tumor for 87.5% of treated animals (Fig. 6b, c). Similarly, αPD-L1 exhibited a modest therapeutic effect on the distant tumor, LID + US

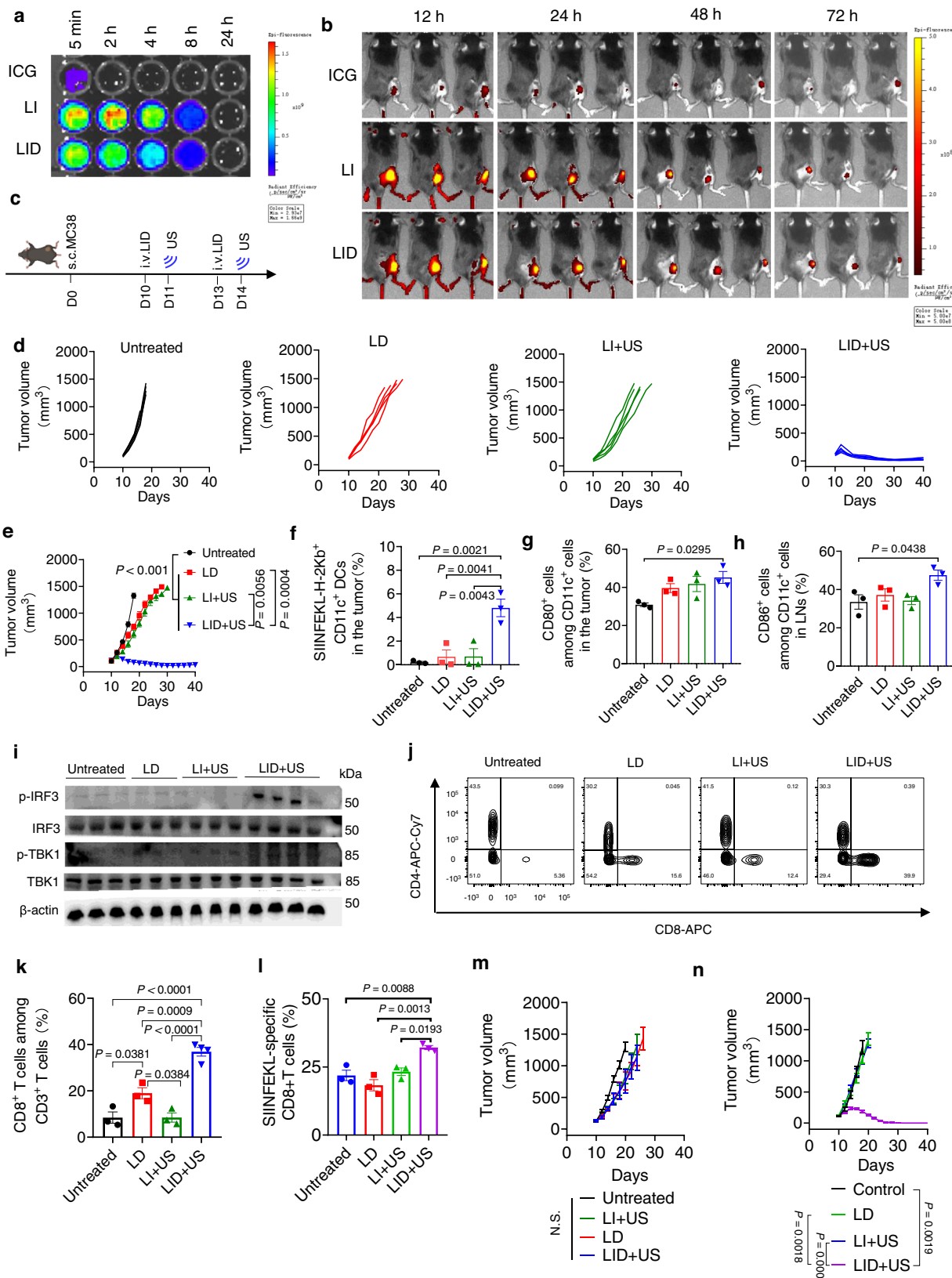

strongly inhibited the distant tumor growth, and LID + US + αPD-L1 regressed the distant tumors for 100% of the bilateral MC38 tumors (Fig. 6d, e). Further analysis revealed that for the primary MC38 tumor, LID + US + αPD-L1 enhanced the intratumoral CD8 + T cell infiltration by 170% and 70% compared with the untreated group (P < 0.05) and LID + US (P < 0.05), respectively (Fig. 6f, g). For the distant MC38

tumor, LID + US + αPD-L1 enhanced the intratumoral CD8 + T cell infiltration by 37% and 70% compared with αPD-L1 (P < 0.01) and LID + US (P < 0.01), respectively, and the latter two groups failed to show any difference compared with the untreated control (Fig. 6h, i). To test if animals have acquired long-term protection, bilateral MC38 tumor-bearing female mice cured by LID + US + αPD-L1 were

**Fig. 5 | Ultrasound responsive chemotherapeutics potently inhibited tumor growth in vivo and induced strong antitumor immunity. a** C57BL/6 mice were intravenously injected with indicated formulations and the blood samples were collected at indicated time points and imaged by the IVIS optical imaging system. Shown are fluorescent images of blood samples at indicated time points (*n* = 3 mice per group). **b** MC38 tumor-bearing mice were intravenously injected with indicated formulations and the animals were imaged by the IVIS optical imaging system at indicated time points (*n* = 3 mice per group). **c** C57BL/6 mice were subcutaneously injected with 500,000 MC38 cells on day 0. On days 10 and 13, tumor-bearing mice were i.v. injected of LID (DOX 0.5 mg/kg, ICG 4 mg/kg) or control formulations. On days 11 and 14, ultrasound (2 W/cm², 50%, 1 MHz, 5 min) was performed for selected groups. **d, e** Individual and average tumor growth curves for MC38 tumor-bearing mice treated with indicated formulations (*n* = 5 mice per group). **f–h** Antigen presentation and DC activation in MC38-OVA tumor-bearing mice two days after ultrasound treatment (*n* = 3 mice per group). **i** Activation of STING pathway-related markers (phosphorylation of IRF3 and TBK1) in the tumor on day 18 following treatment with indicated groups (*n* = 4 mice for LID + US and *n* = 3 mice for all other groups). **j, k** Percent of CD8 + T cells among CD3 + T cells in the tumor microenvironment on day 18 post tumor inoculation in MC38 tumor-bearing mice (*n* = 4 mice for LID + US and *n* = 3 mice for all other groups). **l** Percent of SIINFEKL-specific CD8 + T cells among CD3 + T cells in the tumor on day 18 post tumor inoculation in MC38-OVA tumor-bearing mice (*n* = 3 mice per group). **m** STING KO C57BL/6 mice were subcutaneously injected with 500,000 MC38 cells on day 0 and treated as described in **d**. **n** C57BL/6 mice were subcutaneously injected with 500,000 STING KO MC38 cells on day 0 and treated as described in **d**. Shown are the average tumor growth curves for indicated groups (*n* = 3 mice per group). The data represent mean ± SEM (**e–h, k–n**). Data were analyzed by one-way ANOVA (**f–h, k, l**) or two-way ANOVA (**e, m, n**) with Tukey's multiple comparisons post test. N.S., non-statistically significant. Source data are provided as a Source Data file.

rechallenged by the same MC38 tumor cells on day 40. All treated animals were protected from rechallenge, while the age-matched naive female mice all developed tumors (Supplementary Fig. 23). Similarly, we found LID + US + αPD-L1 also exhibited a more potent therapeutic effect on bilateral CT26 tumors in female mice than LID + US and αPD-L1 (Supplementary Fig. 24), with 60% primary tumors and 80% distant tumors completely regressed by LID + US + αPD-L1. Compared with LID + US and αPD-L1, LID + US + αPD-L1 exhibited a more potent therapeutic effect on primary B16F10 tumors (exposed to ultrasound) and distant B16F10 tumors (not exposed to ultrasound) in female mice (Supplementary Fig. 25). Strikingly, LID + US + αPD-L1 also exhibited a more potent therapeutic effect on clinically relevant orthotopic 4T1 tumors in female mice compared with LID + US and αPD-L1, with 100% primary tumors (exposed to ultrasound) and distant tumors (not exposed to ultrasound) regressed by LID + US + αPD-L1 (Fig. 7a–c). Accordingly, animals receiving LID + US had longer survival than those receiving LD or LI + US (Fig. 7d). Altogether, these results demonstrate that LID + US + αPD-L1 could potently regress established tumors and prevent metastasis and relapse.

## Discussion

In this study, we maximized oxidized endogenous tumor mitochondrial DNA-mediated cGAS-STING activation for cancer immunotherapy by using ultrasound responsive chemotherapeutics LID. Our results indicate that upon exposure to ultrasound, LID containing an optimal ratio of ICG and DOX efficiently generated ROS, which promoted the nuclear delivery of DOX, achieved up to 123-fold increase in tumor cell killing compared with liposomal DOX (LD), and induced mitochondrial DNA oxidation within tumor cells. The enhanced tumor cell killing facilitated the transport of oxidized mitochondrial DNA to APCs, resulting in efficient secretion of type I interferons. We demonstrated that both tumor mitochondrial DNA and STING pathway in APCs were critical in mediating the type I cytokine secretion. Systemic injection of a low-dose LID followed by US treatment over the tumor was able to induce targeted tumor killing and STING activation in the tumor, resulting in potent antitumor T cell immunity, which upon the combination with αPD-L1, regressed multiple types of tumors and protected animals from relapse (Fig. 8).

The fact that chemotherapeutics-killed tumor cells can potentially transport tumor DNA to APCs to activate the STING pathway creates great opportunities to promote antitumor T cell immunity, but chemotherapeutics suffered from poor targeted tumor cell killing, making it difficult to transfer tumor DNA, as well as tumor antigens, to APCs. Furthermore, tumor DNA can be degraded by DNase in APCs and then lose its function to activate the STING pathway. We got the inspiration from phagocytes, which produce ROS to kill pathogens and oxidize pathogen DNA to enhance stability. In fact, hydrogen peroxide has recently been used to inactivate pathogens for the preparation of inactivated vaccines[37,38]. These studies motivated us to utilize ROS to enhance the tumor-killing effect of chemotherapeutics and oxidize tumor DNA to overcome the crucial challenges encountered by chemotherapeutics. We chose to use ultrasound and sonosensitizers to generate ROS as this approach is safe, non-radioactive, and broadly applicable to most parts of the body. Upon exposure to ultrasound, LID efficiently generated ROS, and the presence of chemotherapeutics did not change the level of ROS or intracellular distribution of ROS (Fig. 2a, b). Surprisingly, LID + US promoted the nuclear delivery of DOX compared with LD, although both LID + US and LD had similar levels of total DOX inside cells (Fig. 2c). The enhanced nuclear delivery of DOX achieved by LID + US was largely abrogated in the presence of ROS scavenger, indicating ROS plays an important role in mediating the process. As lysosomal sequestering of nanoparticles is a common issue that can reduce the bioavailability of chemotherapeutics in the nuclei and compromise the therapeutic efficacy, LID + US provides a simple and controllable strategy to overcome this challenge by promoting the endosomal escape of DOX and redistributing DOX to the nuclei. Moreover, ROS also exhibited a direct tumor cell killing effect, thus further contributing to the enhanced tumor-killing effect of LID + US, which had more potent killing of MC38 and CT26 tumors cells compared with LD, a formulation similar to the commercially available Doxil ® (Fig. 3). In fact, chemotherapy with DOX alone typically requires a high dose (5 mg/kg DOX) to induce a meaningful therapeutic effect[39]. In our study, the liposomal DOX (LD) was administered at 0.5 mg/kg, which as chemotherapy alone failed to induce a meaningful therapeutic effect. In contrast, LID + US administered at 0.5 mg/kg of DOX induced a potent therapeutic effect. We demonstrated that this was partially achieved by promoting the delivery of DOX to nuclei of tumor cells, inducing tumor mitochondrial DNA oxidation, and facilitating the transfer of oxidated tumor DNA to APCs to maximize APC activation, which ultimately induced strong antitumor immunity and potent therapeutic effect. Importantly, as we used a low-dose chemotherapeutics and non-tumor regions are not exposed to elevated levels of ROS, our approach can reduce the toxicity on normal tissues compared with traditional chemotherapies while inducing a more targeted cytotoxic effect in the tumor region. Indeed, animals receiving LID + US did not show any sign of side effects. Furthermore, since our strategy is focused on amplifying the cytotoxicity of DOX at the subcellular level by promoting DOX delivery to nuclei and by using ROS to functionally synergize with DOX, it is complementary to previous strategies such as targeting moieties-modified nanoparticles that aim to directly improve the drug delivery at the tissue level and cellular level. Consequently, our strategy is compatible with these active targeting strategies. For example, future studies may incorporate targeting moieties to the ultrasound responsive formulations, which may combine the benefits of directly improving the tumor-targeted delivery and ultrasound-amplified cytotoxic effect in the tumor region.

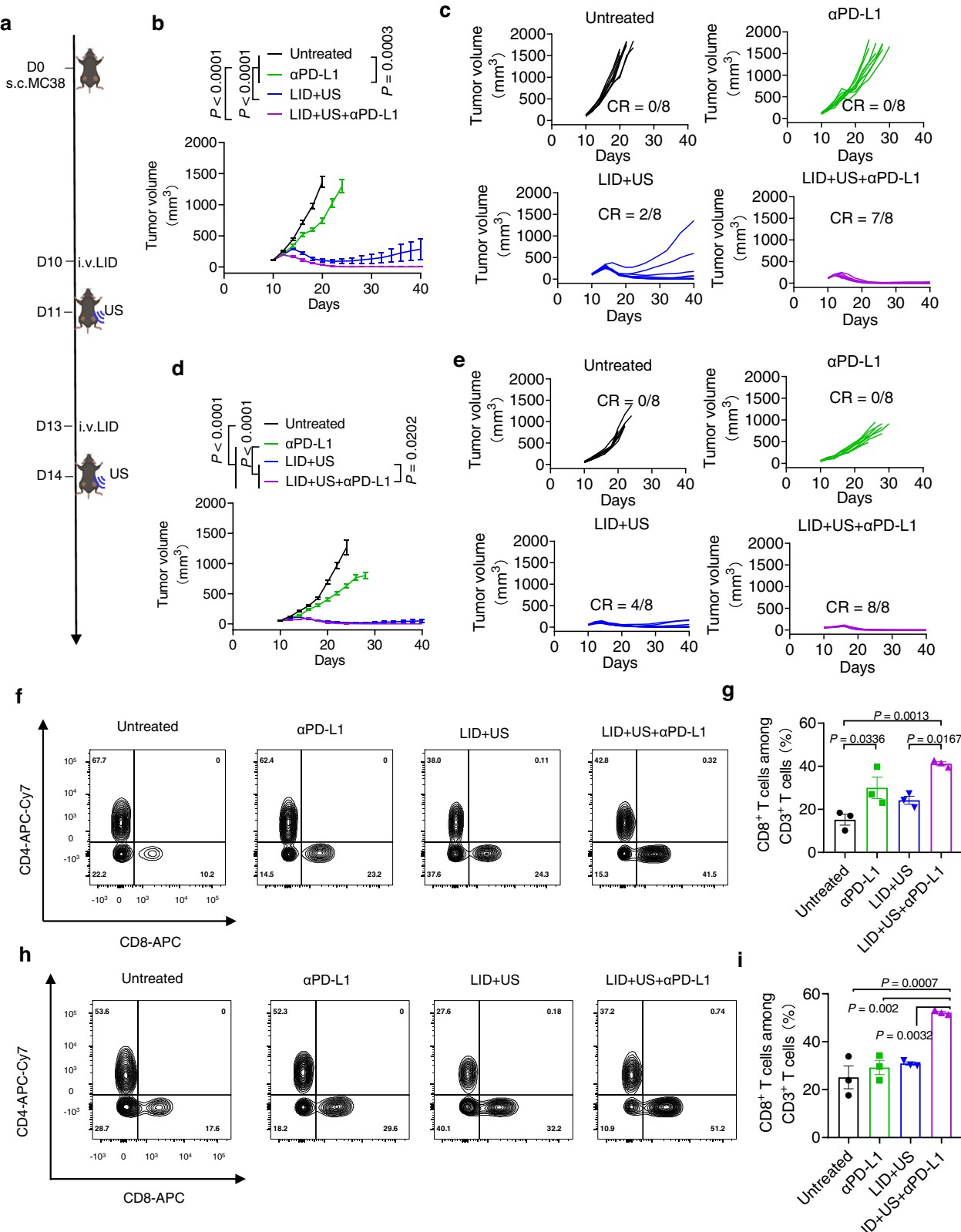

Another critical feature of ultrasound responsive LID is its capability to oxidize tumor DNA. Previous studies have shown that the presence of oxidative modifications such as 8-hydroxyguanosine (8-OHG) and 8-hydroxy-2′-deoxyguanosine (8-OHdG) in oxidized DNA is sufficient to resistant DNase III-mediated degradation and better activate the STING pathway[24]. We found both LID + US and LI + US caused similar levels of tumor DNA oxidation based on the presence of

8-OHdG, and most oxidized tumor DNA was localized in mitochondria (Fig. 4a). We speculate that the distribution pattern of oxidized tumor DNA may reflect that the presence of other barriers such as histones in the nuclei can protect nuclear DNA from oxidation[35]. Interestingly, only LID + US induced significant oxidized tumor DNA transport to APCs, while LD or LI + US failed to achieve this (Fig. 4b). Since LID + US and LI + US induced similar tumor DNA oxidation, these results implied

**Fig. 6 | Ultrasound responsive chemotherapeutics sensitized checkpoint inhibitors on animals with MC38 bilateral tumors. a** C57BL/6 mice were subcutaneously injected with 500,000 MC38 cells on the right flank (primary tumor) and 250,000 MC38 cells on the left flank (distant tumor) on day 0. On days 10 and 13, tumor-bearing mice were i.v. injected with LID (DOX 0.5 mg/kg, ICG 4 mg/kg) or control formulations. On days 11 and 14, ultrasound (2 W/cm², 50%, 1 MHz, 5 min) was applied to the primary tumor for indicated groups (the distant tumor was not exposed to ultrasound). On days 10, 13, and 16, the PD-L1 antibody (75 μg/dose) was i.p. injected for indicated groups. **b**, **c** The average and individual tumor growth curves for primary tumors (exposed to ultrasound) (*n* = 8 mice per group). **d**, **e** The average and individual tumor growth curves for distant tumors (not exposed to ultrasound) (*n* = 8 mice per group). CR = complete regression. **f**, **g** Percent of CD8 + T cells among CD3 + T cells in the primary tumor on day 18 post tumor inoculation. **h**, **i** Percent of CD8 + T cells among CD3 + T cells in the distant tumor on day 18 post tumor inoculation (*n* = 3). The data represent mean ± SEM (**b**, **d**, **g**, **i**). Data were analyzed by one-way ANOVA (**g**, **i**) or two-way ANOVA (**b**, **d**) with Tukey's multiple comparisons post test. Source data are provided as a Source Data file.

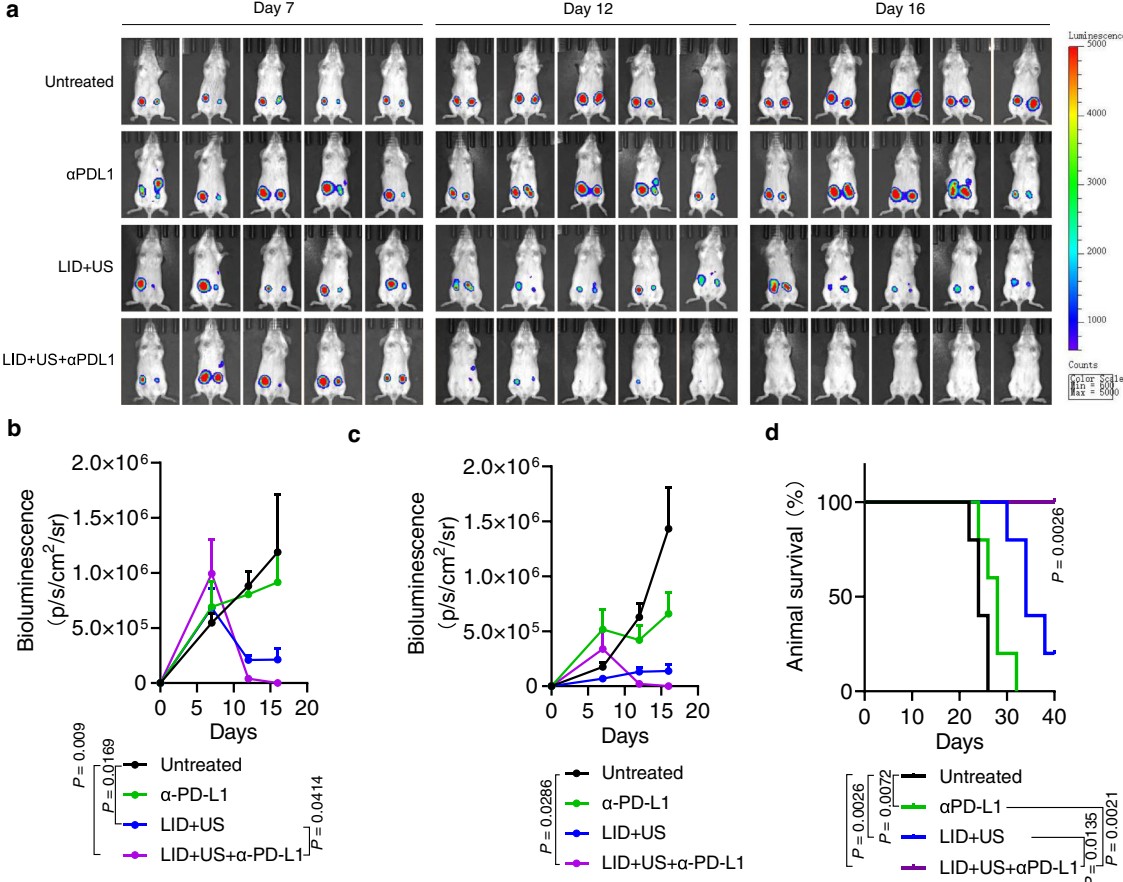

**Fig. 7 | Ultrasound responsive chemotherapeutics sensitized checkpoint inhibitors on animals with orthotopic 4T1 tumors. a** Balb/c mice were injected with 500,000 4T1-Luciferase cells in the right mammary fat pads (primary tumor) and 250,000 4T1-Luciferase cells on the left mammary fat pads (distant tumor) on day 0. On days 7, 10, and 13, tumor-bearing mice were i.v. injected with LID (DOX 0.5 mg/kg, ICG 4 mg/kg) or control formulations. On days 8, 11, and 14, ultrasound (2 W/cm², 50%, 1 MHz, 5 min) was applied to primary tumors for selected groups (distant tumors were not exposed to ultrasound). On days 7, 10, and 13, the PD-L1 antibody (75 μg/dose) was i.p. injected for indicated groups. **b**, **c** The bioluminescence from primary tumors (**b**) and distant tumors (**c**) (*n* = 5 mice per group). **d** Survival of animals treated with indicated formulations. Data were analyzed by two-way ANOVA with Tukey's multiple comparisons post test (**b**, **c**) or log rank (Mantel-Cox) test (**d**). The data represent mean ± SEM (**b**, **c**). Source data are provided as a Source Data file.

that enhanced tumor cell killing induced by LID + US was critical in facilitating the DNA transport to APCs. Consequently, LID + US induced stronger STING activation than LD and LI + US (Fig. 4c, d). Depleting tumor mitochondrial DNA or knocking out STING in APCs significantly compromised type I interferon secretion induced by LID + US, indicating tumor mitochondrial DNA and STING in APCs were indeed critical in mediating the response. However, our data do not exclude the possibility that other DAMPs such as HMGB1 and genomic DNA can mediate activation of APCs. In fact, the presence of residual type I interferon after mitochondrial DNA depletion from tumor cells or STING knockout in APCs implied that other DAMPs from dying tumor cells and PRRs in APCs were also involved in the activation of APCs (Fig. 4d). Furthermore, compared with the use of exogenous agonizts to activate the STING pathway, our approach provides a

simple and effective approach to utilize the abundant tumor DNA to activate the STING pathway in situ and therefore can bypass the limited tumor-targeted delivery of exogenous STING agonists. Future studies are warranted to understand the form and mechanism in which oxidized tumor DNA was transported to APCs.

The potent in vitro activity of LID + US was translated to potent activity in vivo as well. In particular, LID + US enhanced tumor growth inhibition and activation of innate immune cells such as dendritic cells in the tumor and tumor-draining lymph nodes, and promoted infiltration of CD8 + T cells in the tumor compared with LD and LI + US (Fig. 5d–k). The overall tumor growth inhibition effect can be attributed to both the enhanced direct tumor-killing effect and immune activation. Indeed, we have shown that the therapeutic effect of LID + US was compromised in STING knockout mice, indicating the

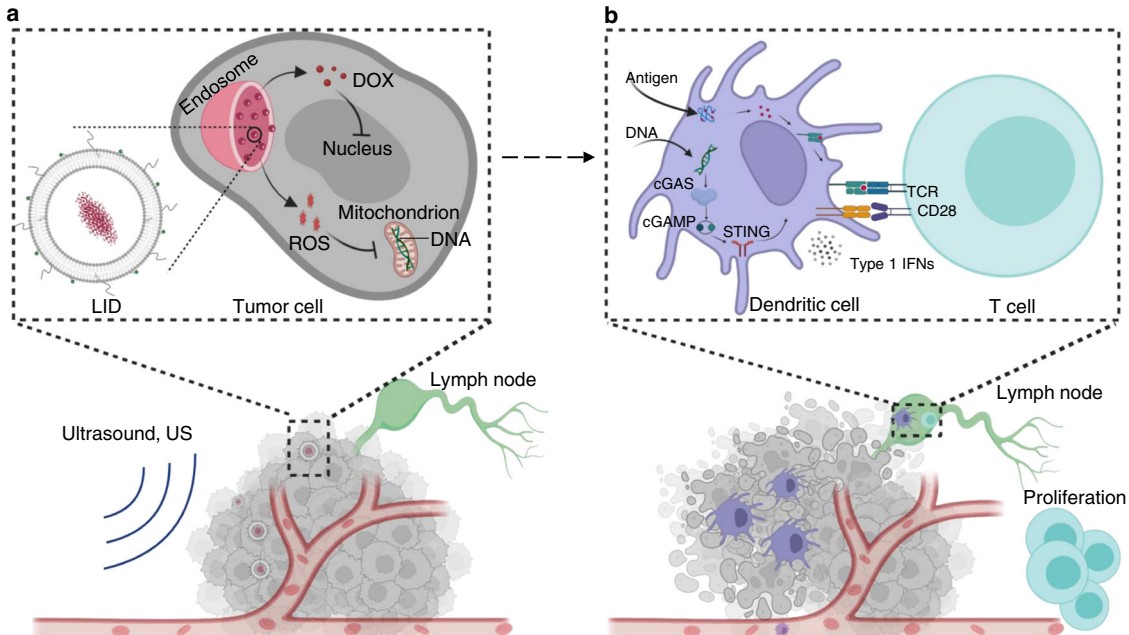

**Fig. 8 | Ultrasound responsive chemotherapeutics for elicitation of potent antitumor T cell immunity. (a)** In the first phase, LID accumulates in the tumor tissue through the enhanced permeability and retention effect (EPR effect). Once the accumulation of LID in the tumor reaches plateau, ultrasound (US) was applied to the tumor region to activate LID to generate ROS, which can significantly enhance DOX delivery to nuclei at the subcellular level and can functionally synergize with DOX to kill tumor cells. Moreover, ROS can also oxidize tumor DNA

(especially mitochondrial DNA) to make it more resistant to nuclease. **(b)** In the second phase, efficient tumor killing induced by LID + US facilitates the transport of tumor antigens and oxidized tumor mitochondrial DNA to tumor infiltrating antigen-presenting cells such as dendritic cells, resulting in enhanced tumor antigen presentation and STING activation, and ultimately activation of potent antitumor T cell immunity that can eliminate remaining tumor cells (not killed during the first phase) and prevent metastasis and relapse.

host STING pathway was indeed critical in mediating the therapeutic effect (Fig. 5m). Depleting CD8 + T cells also compromised the therapeutic effect of LID + US, further confirming the importance of host immune activation (Supplementary Fig. 22). These findings are in line with previous studies that the host STING activation-mediated immunity is important for tumor growth inhibition[7].

In summary, we have developed an ultrasound responsive formulation LID in combination with US to overcome the major challenges encountered by chemotherapeutics to utilize tumor DNA as a DAMP, namely poor transfer of tumor DNA to APCs and poor stability of tumor DNA. Moreover, our study has also advanced the mechanistic understanding of tumor mitochondrial DNA oxidation in STING-mediated antitumor immunity in the context of chemotherapy. The strategy can be potentially applied to other anticancer therapeutics[4] and delivery systems that may benefit from the use of endogenous tumor DNA as an immunosimulatory agent, thus making our approach broadly applicable to the treatment of different tumors.

## Methods
### Ethical statement
This research complies with all relevant ethical regulations. All work performed on animals was in accordance with and approved by the University Committee on Use and Care of Animals at Tsinghua University. Animals were housed in 12 light/12 dark cycle, 65-75 °F (~18-23 °C), and 40-60% humidity condition. Animals were euthanized when the tumor reached 15 mm in any dimension or when they became moribund with severe weight loss or unhealing ulceration. This limit was not exceeded at any point.

### Preparation and characterization of LID
Liposomal ICG/DOX (LID) was prepared by loading DOX in the aqueous core and loading ICG on the surface of preformed liposomes. Briefly, 9.58 mg 1,2-dipalmitoyl-sn-glycero-3-phosphocholine, 3.19 mg

cholesterol, and 3.19 mg 1,2-distearoyl-sn-glycero-3-phosphoethano-lamine-N-[methoxy(polyethylene glycol)−2000] were dissolved in 0.1 ml ethanol, which was then mixed with 0.9 ml 250 mM $(NH_4)_2SO_4$ under magnetic stirring. The mixture was extruded 21 times through a 100-nm polycarbonate membrane (Avanti Polar Lipids) at 50 °C. The obtained blank liposomes were passed through a PD-10 column (GE Healthcare) to remove ethanol and external $(NH_4)_2SO_4$, followed by incubation with 0.55 mg/ml DOX for 30 min at 55 °C to enable the $(NH_4)_2SO_4$ gradient-driven active drug loading. To load ICG, DOPE was firstly conjugated to ICG-NHS to obtain DOPE-ICG. Different concentrations of DOPE-ICG (containing 4.2 mg/ml, 2.1 mg/ml, 1.05 mg/ml, or 0.53 mg/ml ICG) were then incubated with DOX-loaded liposomes for 30 min at room temperature to obtain LID with different ICG/DOX ratios. Unloaded doxorubicin and ICG were removed by passing the liposomes through the PD-10 column. Liposomes containing only ICG or DOX alone were prepared in the same way, with the second drug omitted. The encapsulation efficiency of ICG and doxorubicin was quantified by measuring the fluorescence of ICG (Ex = 751 nm, Em = 830 nm) and DOX (Ex = 485 nm, Em = 590 nm) using a microplate reader (BioTek). The size and zeta potential were measured by dynamic laser scattering (DLS) on a Malvern Zetasizer and the data analysis was performed using Zetasizer software version 7.13. The morphology of liposomes containing ICG and DOX was observed by cryo-electron microscopy (cryo-EM) and cryo-electron tomography (cryo-ET). For cryo-EM, 3 µl sample (10-fold dilution from the original samples) was dropped on a glow discharged copper grid coated with holey carbon (R 2/2; Quantifoil, Jena, Germany), incubated for 1 min, blotted for 3.5-4.5 s, and then plunged into liquid ethane using a Vitrobot Mark IV cryo-sample plunger (Thermo Fisher Scientific, Hillsboro, OR). The samples were loaded on a 200 kV FEI Talos Arctica transmission electron microscope (Thermo Fisher Scientific, Hillsboro, OR) equipped with a Ceta-D camera. Images were recorded at nominal magnifications of 73,000× or 92,000× and at a defocus of −3.5 µm. For

cryo-ET imaging, 3 μl sample was mixed with gold fiducial beads (10 nm diameter, Aurion, the Netherlands), plunged with Vitrobot Mark IV and imaged on a 200 kV FEI Talos Arctica transmission electron microscope equipped with a K2 direct electron detector (Gatan, CA). Samples were recorded at a magnification of 28,000×, resulting in a pixel size of 1.51 Å. Tilt-series data were collected using the bidirectional scheme from 0 to −60, and +3 to +60°, or from +15 to −60, and +18 to +60°, at 3° steps at a defocus of −3.5 or −3.8 μm in SerialEM[40]. For each tilt series, a movie consisting of 8 frames was recorded with an exposure time of 0.1s/frame. The electron beam-induced motion among each tilt was corrected by MotionCor2[41]. Tomograms were reconstructed by weighted back projection in IMOD[42], 4× binned and lowpassed to 80 Å resolution. The movies from multiple tomogram slices were generated through ImageJ[43] with scale bars of 50 nm. To learn the stability of LID, 500 μl LID containing 400 μg/ml ICG and 50 μg/ml DOX was added into a dialysis tube (MWCO = 1000 kDa) and incubated in 15 ml PBS or PBS containing 10% FBS at 37 °C under 130 rpm constant shaking. Samples outside the dialysis tube were collected at predetermined time points over 48 h and the concentration of DOX and ICG were quantified as described above. To learn the release profile of DOX and ICG under different conditions, 500 μl LID containing 400 μg/ml ICG and 50 μg/ml DOX was mixed with 4.5 ml release media (pH 4.5 citric acid buffer or pH 7.4 PBS) and incubated at 37 °C under 130 rpm constant shaking. Ultrasound (2 W/cm², 50% duty cycle, 1 MHz, 5 min) was applied to indicated groups at the 24 h time point. Samples (120 μl) were collected at predetermined time points and passed through Zeba™ Spin Desalting Columns (7 K MWCO) to separate released drug from liposomes. The drug concentrations in liposomes were quantified by measuring the fluorescence of ICG and DOX as described above.

## ROS generation and intracellular DOX delivery

Throughout the studies, all cells were tested negative for mycoplasma contamination and morphologically confirmed. To quantify ROS generation, 10,000 MC38 or CT26 cells were incubated with indicated formulations containing 3 μM ICG or 0.5 μM DOX for 24 h at 37 °C. Then the old medium was replaced with fresh medium containing 2′,7′-Dichlorodihydrofluorescein diacetate (DCFH-DA, 1:2000 dilution). Ultrasound (2 W/cm², 50% duty cycle, 1 MHz) was applied to selected groups for indicated lengths of time. After incubation for 20 min at 37 °C, the fluorescence was measured on a microplate reader (BioTek). To image ROS generation in cells, 50,000 MC38 or CT26 cells were incubated with indicated formulations containing 3 μM ICG or 0.5 μM DOX for 24 h at 37 °C. Then the old medium was replaced with fresh medium containing DCFH-DA (1:2000 dilution). Ultrasound (2 W/cm², 50% duty cycle, 1 MHz, 5 min) was applied to selected groups. After incubation for 20 min at 37 °C, the cells were washed with PBS, stained with DAPI, and then imaged by a confocal microscope (Zeiss LSM780). To image intracellular DOX delivery, 50,000 MC38 tumor cells were incubated with indicated formulations containing 3 μM ICG or 0.5 μM DOX for 24 h at 37 °C. Then the old medium was replaced with fresh medium. Ultrasound (2 W/cm², 50% duty cycle, 1 MHz, 5 min) was applied to selected groups. After incubation for 24 h at 37 °C, the cells were washed with PBS, stained with DAPI, and then imaged by a confocal microscope (Zeiss LSM780). γ-H2AX was detected using the Beyotime's DNA damage Assay Kit (γ-H2AX Immunofluorescence) following the manufacturer's instruction. To measure the endosomal membrane permeability, 50,000 MC38 tumor cells were treated with LI containing 3 μM ICG for 24 h. Ultrasound (2 W/cm², 50% duty cycle, 5 min) was applied to the selected group (LI + US). Acridine Orange (MedChemExpress, 5 μM) was added immediately after ultrasound treatment and incubated for 30 min at 37°C before imaging by a confocal microscope (Zeiss LSM780).

## Measurement of cytotoxicity and ICD markers

To measure the cytotoxicity of different formulations on tumor cells, 5000 MC38 or CT26 tumor cells were incubated with indicated formulations containing predetermined concentrations of ICG and DOX for 24 h at 37 °C. Then the old medium was replaced with fresh medium, and ultrasound (2 W/cm², 50% duty cycle, 1 MHz) was applied to selected groups. In some experiments, the ultrasound time was varied between 1 min and 7.5 min. After incubation for 24 h at 37 °C, the cell viability was measured using the cell counting kit-8 (GLPBIO) following the manufacturer's instructions. To analyze the levels of immunogenic cell death markers such as calreticulin (CRT, also called calregulin) and high-mobility group box 1 (HMGB1), 200,000 MC38 tumor cells were treated with indicated formulations containing 3 μM ICG and/or 0.5 μM DOX for 24 h. Ultrasound (2 W/cm², 50% duty cycle, 5 min) was applied to selected groups. After 24 h, cells were washed with FACS buffer, followed by incubation with CD16/32 (1:20) and then incubation with Calregulin (F-4) Alexa Fluor® 488 (1:100) for 30 min before flow cytometry. HMGB1 levels in the supernatant were measured using the ELISA kit following the manufacturer's instructions.

## DNA oxidation and transport

To monitor the DNA oxidation in tumor cells, 50,000 MC38 or CT26 tumor cells were treated with indicated formulations containing 3 μM ICG and/or 0.5 μM DOX for 24 h. Then the old medium was replaced with fresh medium, and ultrasound (2 W/cm², 50% duty cycle, 1 MHz, 5 min) was applied to selected groups. After culturing for 24 h at 37 °C, the cells were incubated with MitoTracker to label the mitochondria and incubated with an anti-8OHdG antibody (1:50) to label the oxidized tumor DNA following the manufacturer's instructions. The cells were washed with PBS, stained with DAPI to label the nuclei, and then imaged by a confocal microscope (Zeiss LSM780). To monitor oxidized tumor DNA transport, 50,000 MC38 or CT26 tumor cells were treated with indicated formulations containing 3 μM ICG and/or 0.5 μM DOX for 24 h. Then the old medium was replaced with fresh medium, and ultrasound (2 W/cm², 50% duty cycle, 1 MHz, 5 min) was applied to selected groups. After culturing for 24 h at 37 °C, 50,000 BMDCs were added and incubated for another 24 h. The cells were incubated with anti-CD11c and anti-8OHdG to label the dendritic cells and oxidized tumor DNA, respectively. The cells were washed with PBS, stained with DAPI to label the nuclei, and then imaged by a confocal microscope (Zeiss LSM780).

## STING activation in antigen-presenting cells

To measure activation of RAW-lucia™ ISG reporter cells, 200,000 MC38-OVA or CT26 tumor cells were treated with indicated formulations containing 3 μM ICG and/or 0.5 μM DOX for 24 h. Then the old medium was replaced with fresh medium, and ultrasound (2 W/cm², 50% duty cycle, 1 MHz, 1–5 min) was applied to selected groups. After culturing for 24 h at 37 °C, 200,000 RAW-lucia™ ISG reporter cells were added and incubated for another 24 h. The activation of RAW-lucia™ ISG reporter cells was quantified by mearing the luminescence on a microplate reader (BioTek) following the manufacturer's instructions. To measure type I interferon (e.g., IFNβ) secretion from primary BMDCs, 200,000 MC38-OVA or CT26 tumor cells were first treated with indicated formulations containing 3 μM ICG and/or 0.5 μM DOX for 24 h. Then the old medium was replaced with fresh medium, and ultrasound (2 W/cm², 50% duty cycle, 1 MHz, 5 min) was applied to selected groups. After culturing for 24 h at 37 °C, 200,000 BMDCs were added and incubated for another 24 h. IFNβ levels in the supernatant were measured by ELISA (Invivogen) following the manufacturer's instructions. In some experiments, mitochondrial DNA in tumor cells was depleted by using 150 μM dideoxycytidine (ddc) to treat tumor cells for 6 days[35] before treatment with indicated formulations.

## Antigen presentation on dendritic cells

Dying tumor cells-induced antigen presentation on dendritic cells was measured following previously established protocols with slight modifications[44]. Briefly, 200,000 MC38-OVA cells were incubated with indicated formulations containing 3 µM ICG and/or 0.5 µM DOX for 24 h. Then the old medium was replaced with fresh medium, and ultrasound (2 W/cm², 50% duty cycle, 1 MHz, 5 min) was applied to selected groups. After incubation for 24 h at 37 °C, 200,000 bone marrow-derived dendritic cells (BMDCs) were added and cultured for another 24 h. BMDCs were collected, washed with FACS buffer, incubated with anti-CD16/32 (1:20) at room temperature for 10 min, and then stained with APC-conjugated anti-mouse SIINFEKL/H-2K$^b$ antibody 25-D1.16 (1:100). Cells were then washed with FACS buffer and resuspended in 2 µg/ml DAPI before analysis by flow cytometry (BD LSRFortessa SORP).

## Pharmacokinetics and biodistribution

For the pharmacokinetic study, female C57BL/6 mice of age 6-8 weeks (Vital river) were intravenously injected with indicated formulations containing 0.5 mg/kg DOX and/or 4 mg/kg ICG. At predetermined time points, the blood samples were collected by retro-orbital bleeding. The fluorescence of ICG was observed by using the IVIS optical imaging system. For the biodistribution study, 500,000 MC38 tumor cells were subcutaneously inoculated on the right flank of female C57BL/6 mice on day 0. On day 10, the tumor-bearing mice were intravenously injected with indicated formulations containing 0.5 mg/kg DOX and/or 4 mg/kg ICG. At predetermined time points, fluorescence of ICG was imaged using the IVIS optical imaging system. The average radiant efficiency was quantitated by IVIS Lumina Living Image Software (v.4.5.5).

## Western blotting

In vitro cell samples or in vivo tumor tissues harvested on indicated days were lysed using the radioimmunoprecipitation assay (RIPA) buffer in the presence of protease and phosphatase inhibitor cocktail. The protein concentration was quantified using Pierce BCA protein kit (Thermo Fisher Scientific). Equal amounts of samples (20 µg) were separated using the SDS-polyacrylamide gel and electro-transferred to polyvinylidene fluoride membranes. Membranes were blocked with 5% non-fat milk in tris-buffered saline containing 0.1% Tween 20 before incubation at 4 °C overnight with the following primary antibodies: β-actin (D6A8, 1:1000), STING/TMEM173 (1:1000), TBK1 (D1B4, 1:1000), phospho-TBK1 (Ser172, 1:1000), IRF3 (D83B9, 1:1000), and phospho-IRF3 (S396, 1:1000). The membranes were further incubated with the goat-anti-rabbit-HRP secondary antibody (whole molecule, 1:1000) and ECL substrate before visualization using a chemiluminescence image analysis system (Tanon 4600).

## Therapeutic study on tumor-bearing mice

For studies with right flank MC38 tumors, female C57BL/6 mice of age 6−8 weeks (Vital river) were subcutaneously inoculated with 500,000 MC38 tumor cells on the right flank on day 0. On days 10 and 13, tumor-bearing mice were intravenously injected with indicated formulations containing 0.5 mg/kg DOX and/or 4 mg/kg ICG. On days 11 and 14, ultrasound (2 W/cm², 50% duty cycle, 1 MHz, 5 min) was applied to the right flank tumor for selected groups. Tumor size was monitored every 2 days, and the tumor volume was calculated by the following equation[45]: tumor volume = length × width$^2$ × 0.52. Animals were euthanized when the individual tumor reached 15 mm in any dimension or when animals became moribund with severe weight loss or had active ulceration. In some experiments, STING knockout female C57BL/6 mice of age 6-8 weeks (GemPharmatech) were subcutaneously inoculated with 500,000 WT MC38 tumor cells on the right flank or female C57BL/6 mice of age 6-8 weeks (Vital river) were subcutaneously inoculated with 500,000 STING KO MC38 tumor cells on the right flank to establish the tumor models and tumor-bearing

mice were treated as described above. For studies with bilateral MC38 tumors, female C57BL/6 mice were subcutaneously inoculated with 500,000 MC38 tumor cells on the right flank and 250,000 MC38 tumor cells on the left flank on day 0. On days 10 and 13, tumor-bearing mice were intravenously injected with indicated formulations containing 0.5 mg/kg DOX and/or 4 mg/kg ICG. On days 11 and 14, ultrasound (2 W/cm², 50% duty cycle, 1 MHz, 5 min) was applied to the right flank tumor for selected groups. On days 10, 13, and 16, the PD-L1 antibody (75 µg/dose) was intraperitoneally (i.p.) injected for indicated groups. The tumor growth was monitored as described above. On day 40, the mice with tumors completely regressed were rechallenged by subcutaneous injection of 500,000 MC38 tumor cells, and age-matched naïve mice without treatment were used as the control group.

For studies with right flank CT26 tumors, female Balb/c mice of age 6-8 weeks (Vital river) were subcutaneously inoculated with 200,000 CT26 tumor cells on the right flank on day 0. On days 10, 13, and 16, tumor-bearing mice were intravenously injected with indicated formulations containing 0.5 mg/kg DOX and/or 4 mg/kg ICG. On days 11, 14, and 17, ultrasound (2 W/cm², 50% duty cycle, 1 MHz, 5 min) was applied to the right flank tumor for selected groups. For studies with bilateral CT26 tumors, female Balb/c mice were subcutaneously inoculated with 200,000 CT26 tumor cells on the right flank and 100,000 CT26 tumor cells on the left flank on day 0. On days 10, 13, and 16, tumor-bearing mice were intravenously injected with indicated formulations containing 0.5 mg/kg DOX and/or 4 mg/kg ICG. On days 11, 14, and 17, ultrasound (2 W/cm², 50% duty cycle, 1 MHz, 5 min) was applied to the right flank tumor for selected groups. On days 10, 13, and 16, the PD-L1 antibody (75 µg/dose) was i.p. injected for indicated groups. The tumor growth was monitored as described above.

For studies with B16F10 tumors, female C57BL/6 mice were subcutaneously injected with 200,000 B16F10 cells on the right flank on day 0 and 100,000 B16F10 cells on the left flank on day 3. On days 8, 11, and 14, tumor-bearing mice were i.v. injected with LID (DOX 0.5 mg/kg, ICG 4 mg/kg) or control formulations. On days 9, 12, and 15, ultrasound (2 W/cm², 50%, 1 MHz, 5 min) was performed for selected groups. On days 8, 11, 14, and 17 the PD-L1 antibody (75 µg/dose) was i.p. injected for indicated groups. The tumor growth was monitored as described above.

For studies with orthotopic 4T1 tumors, female Balb/c mice were injected with 500,000 4T1-Luc cells in the right mammary fat pads (the primary tumor) and 250,000 4T1-Luc cells on the left mammary fat pads (distant tumor) on day 0. On days 7, 10, and 13, tumor-bearing mice were i.v. injected with LID (DOX 0.5 mg/kg, ICG 4 mg/kg) or control formulations. On days 8, 11, and 14, ultrasound (2 W/cm², 50%, 1 MHz, 5 min) was applied to primary tumors for selected groups (distant tumors were not exposed to ultrasound). On days 7, 10, and 13, the PD-L1 antibody (75 ug/dose) was i.p. injected for indicated groups.

For a subset of studies, tumor tissues and tumor-draining lymph nodes were harvested from tumor-bearing mice on indicated days and cut into small pieces, followed by dissociation using the digestion buffer (1 mg/ml collagenase and 100 µg/ml deoxyribonuclease I in serum-free RPMI) for 30 min at 37°C with gentle shaking. The obtained suspension was passed through a 70-µm strainer to obtain the single-cell suspension[45]. The cells were incubated with CD16/32 (1:20) for 10 min and then incubated with anti-CD3 (30-F11) (1:100), anti-CD8 (53.6.7) (1:100), anti-CD11c (N418) (1:100), anti-CD80 (16-10A1) (1:100), anti-CD86 (M5/114.15.2) (1:100), anti-CD4 (GK1.5) (1:100), anti-mouse SIINFEKL/H-2K$^b$ antibody 25-D1.16 (1:100), and peptide-MHC tetramer tagged with PE (H-2K$^b$-restricted SIINFEKL) (1:40) before flow cytometry (BD LSRFortessa SORP). Flow cytometric data were collected using BD FACSDiva Software v8.0.

## Statistical analysis

Sample sizes were chosen based on preliminary data and previously published results in the literature. All animal studies were performed after randomization. Data were analyzed by one-way or two-way

ANOVA, followed by Tukey's multiple comparisons post test with Prism 8.0 (GraphPad Software). P values less than 0.05 were considered statistically significant. All values are reported as means ± SEM unless specified otherwise.

## Reporting summary

Further information on research design is available in the Nature Portfolio Reporting Summary linked to this article.

## Data availability

The authors declare that data supporting the findings of this study are available within the article, Supplementary, or Source data files. RNA-sequencing datasets have been deposited to NCBI-Sequence Read Archive (SRA) under accession code SRP439937. The link of this project is https://www.ncbi.nlm.nih.gov/bioproject/PRJNA977031. Source data are provided with this paper.

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

## Acknowledgements

The work was supported in part by grants from National Natural Science Foundation of China (82173751 to R.K.; 32241031 and 32171195 to S.L.), Tsinghua University Initiative Scientific Research Program (2023Z11DSZ001 to R.K.), start-up packages from Tsinghua University to R.K., support from Tsinghua-Peking Center for Life Sciences to R.K., support from the Key Laboratory of Innovative Drug Research and Evaluation to R.K, and Tsinghua University Spring Breeze Fund to S.L. (2021Z99CFZ004). We thank the Tsinghua University Branch of China National Center for Protein Sciences (Beijing) for the CryoEM facility.

## Author contributions

C.W., R.Z., J.H., and R.K. designed the experiments. C.W., R.Z., J.H., L.Y., and X. L. performed the experiments. C.W., R.Z., J.H., and R.K. analyzed the data. J.Z. and S.L. contributed to the cryo-electron microscopy (cryo-EM) and cryo-electron tomography (cryo-ET). C. Z, J.C.K, and J.M.K. contributed to data analysis and critically reviewing and editing the manuscript. C.W. and R.K. wrote the manuscript. We acknowledge James Moon (University of Michigan) for critical review of the manuscript prior to submission. All authors have given approval to the final version of the manuscript.

## Competing interests

A patent application (202210306460.1) has been filed based on the ultrasound responsive chemotherapeutics for STING activation, with R.K., C.W., R.Z., and J.H. as inventors. J.M.K. and R.K. hold equity and consult for Corner Therapeutics, a company that has licensed IP generated by J.M.K. and R.K. that may benefit financially if the IP is further validated. J.M.K. has been a paid consultant and or equity holder for multiple biotechnology companies (listed here https://www.karplab.net/team/jeff-karp). The interests of JMK were reviewed and are subject to a management plan overseen by his institutions in accordance with its conflict of interest policies. The remaining authors declare no competing interests.
