## [Peer Review File · Nature Communications]

Reviewers' Comments:

Reviewer #1:

Remarks to the Author:

Line 53 - yet, you expect that APCs will penetrate the inner recesses of a tumour and have better tumour cell kill? Explain how this may work if smaller molecules cannot penetrate deep?

Line 89 - Should read 'chemotherapeutic'.

Line 164 - did you try higher ratios than 8:1 for your LID formulation? If not, then why not, as the trend would be that you would get more anticancer effect?

Line 173 - did you try > 5min for US time? Comment on the cell kill ability increase with increasing time for LI formulation.

Line 180 - why only use CRC cell lines, and only two, and none of which are human? To show impact, at least 3 different types of cancer types, with 2 cell lines of each, and at least one of each type being of human origin, would be the minimum starting point.

Line 191 - supply citation of histones in nuclei being able to protect nDNA, if not, it is purely speculation, and should be clearly stated as such.

Line 224 - it would be better if a natural non-forcibly expressed protein was used as epitope to prove this point. Data would be more translationally relevant.

Line 293 - without toxicity/safety data, this dataset and the manuscript therefore is not acceptable. Note that for a drug like Dox, which is highly effective in killing tumor cells, it is the toxicity to normal tissues that forces the chief oncologist to reduce or cease further treatment. Furthermore, I would like to emphasise that in an 'acute' (for want of a better term) model like this, where animals were dosed for 40 days, the accumulation of drug is not allowed to occur to the point where it becomes toxic, even if treatment is ceased. Dox is not only toxic during dosing, but months or years after due to its ability to accumulate in the body. It is not enough to show efficacy data alone, as lots of promising anticancer technologies stumble later on due to the lack of toxicity/safety data at the outset. Thus, it is imperative that toxicity data is shown.

Line 293 - further to the above, it would have helped greatly if normal cells were also exposed to the technology described. Cardiomyocytes such as differentiated H9c2 would have been a good model. Note that this would do away with the need for in vivo safety evaluation, but would complement it.

Line 481 - the sc model chosen here is too simplistic and data therefore hardly clinically relevant. For instance, a sc model does not incorporate any element of metastasis, but is one solid tumour, at the wrong spot (ie. away from the colorectal area), when compared to a more clinically-relevant orthotopic model which would incorporate cells being shed from the site of injection to neighboring tissues such as liver and stomach. To put it in a different way, a primary tumor like the one established in this study would be easy to resect if presented clinically, so Dox treatment would not be critical or even required. As such, there is a flaw in the design of this study, and at the most critical point - when the team is trying to prove that their technology works. Yes, it does work, rather well, in this oversimplified model. But, is it clinically-facing or -relevant? I doubt that.

Reviewer #2:

Remarks to the Author:

This manuscript describes an interesting approach to targeting liposomal doxorubicin/indocyanine green to tumors using ultrasound. They demonstrate enhanced effect over liposomal doxorubicin and demonstrate immune activation and synergy with an anti-PD1 treatment in a murine model. The approach may have some therapeutic advantages in allowing the specific targeting of tumors. There are, however, some issues that need addressed.

Issues:

Fig 3a/b: These would benefit from a higher magnification image. How was intracellular doxorubicin imaged? Was an antibody used?

Figs 3c-h: An appropriate control is missing. What effect did LID have without ultrasound? This is important as the authors noted that the acidic pH in endosomes could trigger DOX release from LID without ultrasound treatment.

Fig 4. The authors have only focused on oxidised DNA damage as the main mechanism of action. Doxorubicin is a potent topoisomerase II inhibitor and induces double stranded DNA breaks. They need to look at markers of DNA breaks in the nucleus such as H2AX foci. This will likely demonstrate nuclear DNA damage. This is important as nuclear DNA damage leads to the accumulation of cytosolic DNA in tumors thereby triggering the cGAS-STING pathway. (Harding et al Nature. 2017 Aug 24; 548(7668): 466-470). This is independent on APCs.

Fig 4e: There needs to be more detail on how mitochondria were depleted in the manuscript other than a mention of ddC in the figure legend. These data do not necessarily support oxidised mitochondrial DNA as the main activator of cGAS-STING. It is equally possible that loss of mitochondrial function has an indirect negative effect on STING/interferon signaling. Does treatment with exogenous cGAMP still activate the STING pathway in the absence of mitochondria? Do DNA damaging agents that target nuclear DNA (such as etoposide) still activate cGAS-STING?

Fig 5: The authors describe the RAW-lucia ISG cell line as a reporter for cGAS/STING activation. This is inaccurate. It is a reporter for interferon alpha and beta release following potential activation of several pattern recognition receptors. It is not specific to cGAS/STING activation.

Fig 5i. It is unclear where STING activation is occurring in response to the treatment. Is it in the cancer cells or dendritic cells or both? As mentioned above there is good evidence that DNA damaging agents activate cGAS-STING directly in tumors resulting in cytokine release and T cell recruitment (this is independent of APCs). The authors therefore need to look at cGAS-STING (IRF3 and TBK1 activation) following LID/US treatment in MC38 cells in-vitro. STING should then be knocked out of this cell line to show dependence.

Fig 6: The authors need to explain if in the bilateral tumor experiments both tumors were treated with US. It would be more clinically relevant if metastatic sites regressed following treatment of the primary tumor only, as often metastatic sites are either inaccessible or too numerous to treat directly.

Reviewer #3:

Remarks to the Author:

This contribution describes a study of liposomes containing both indocyanine green as a 'sonosensitizer' and doxorubicin as an anti-tumor agent and their ability to inhibit tumor cell growth in culture and in xenograft models when exposed to ultrasound. Although there are extensive experiments performed showing modest tumor suppression cell culture and more significant tumor remissions in vivo, there are two major flaws in this reviewer's view that make this work unpublishable in its current form.

1. The mode of action of sonosensitization is poorly understood in general and this contribution does nothing to improve our understanding of the physical basis of ROS generation. Although there are data presented showing modest biological impacts as a function of sonoexcitation time, there are no data presented showing how the DOPE-ICG loading impacts the function of the liposome formulations on cells. Since the in vitro experiments are conducted with 24 hour exposures of the cells to the liposome formulations, there is ample time for the agents to be internalized by the cells and repurposed to other organelles prior to sonoexcitation. It is not clear that the ROS production is a result of the intact formulation or cellular redistribution of the liposomal components. Higher resolution and time-dependent studies are needed to better understand this system.

2. There is an unacceptable lack of an important control throughout the study -- that is, the impact of liposomal doxorubicin + US in the absence of ICG. It is conceivable that the nanocrystalline DOX is sonochemically active by itself and is responsible for the reported effects by altering its dissolution profile once in the cell and/or tumor tissue environment in vivo. The LI + US control

was reported, however, a LD + US control must be included in the study to more clearly demonstrate the necessity of including ICG in the formulation to achieve the reported bioactivity.

We highly appreciate the reviewers' thorough review of our manuscript and their constructive comments. We have carefully addressed the reviewers' concerns in the following point-by-point responses and revised the manuscript accordingly.

Reviewers' comments:

Reviewer #1 - Chemotherapy, mitochondrial stress (Remarks to the Author):

Line 53 - yet, you expect that APCs will penetrate the inner recesses of a tumour and have better tumour cell kill? Explain how this may work if smaller molecules cannot penetrate deep?

We thank the reviewer for the constructive comments. The mechanism of action for LID+US is shown in Figure 1. (a) In the first phase, LID accumulates in the tumor tissue through the enhanced permeability and retention effect (EPR effect). Once the accumulation of LID in the tumor reaches plateau, ultrasound (US) was applied to the tumor region to activate LID to generate ROS, which can significantly enhance DOX delivery to nuclei at the subcellular level and can functionally synergize with DOX to kill tumor cells. Moreover, ROS can also oxidize tumor DNA (especially mitochondrial DNA) to make it more resistant to nuclease. (b) In the second phase, efficient tumor killing induced by LID+US facilitates the transport of tumor antigens and oxidized tumor mitochondrial DNA to tumor-infiltrating antigen presenting cells (APCs) such as dendritic cells, resulting in enhanced tumor antigen presentation and STING activation, and ultimately activation of potent antitumor T cell immunity that can eliminate remaining tumor cells (not killed during the first phase) and prevent metastasis and relapse.

While numerous studies have shown that tumor-infiltrating APCs can penetrate the tumor and acquire tumor antigens and DAMPs from dying tumor cells to activate the antitumor immunity (*Nat Commun.* 2016;7:12499. PMID: 27530650; *Nat Commun.* 2019;10(1):2025. PMID:31048681), our approach is the first report that focuses on amplifying the cytotoxicity of low-dose DOX at the subcellular level by promoting DOX delivery to nuclei, and by using ROS to functionally synergize with DOX to kill tumor cells. Coupled with the efficient use of oxidized tumor mitochondrial DNA to activate the STING pathway in APCs for elicitation of antitumor immunity, our strategy has potent antitumor efficacy in vivo. Furthermore, since our strategy is complementary to previous strategies such as active targeting moieties-modified nanoparticles that aim to directly improve the drug delivery at the tissue level (e.g., improving accumulation and penetration) and cellular level (e.g., improving cellular uptake), which typically suffer from lysosomal sequestering of nanoparticles and poor DOX delivery to the nuclei, our strategy is compatible with these active targeting strategies and may have broad applications in the future.

Line 89 - Should read 'chemotherapeutic'.

We have updated the text.

Line 164 - did you try higher ratios than 8:1 for your LID formulation? If not, then why not, as the trend would be that you would get more anticancer effect?

A ratio higher than 8:1 is over the loading capacity of liposomes and therefore not tested.

Line 173 - did you try > 5min for US time? Comment on the cell kill ability increase with increasing time for LI formulation.

The tumor cell killing effect of LI was mainly due to the US-triggered generation of cytotoxic ROS, whose levels were improved with the increase of US time (Fig.3a), so the tumor killing ability of LI was positively correlated with ultrasound time within a certain range. We have shown that the tumor killing effect of LI+US was much weaker than LID+US, indicating the presence of DOX was critical in enhancing the tumor killing effect. Further increasing the ultrasound time from 5 min to 7.5 min did not significantly enhance the tumor cell killing of LID+US (Fig.4b-c), so we used 5 min for most experiments.

Line 180 - why only use CRC cell lines, and only two, and none of which are human? To show impact, at least 3 different types of cancer types, with 2 cell lines of each, and at least one of each type being of human origin, would be the minimum starting point.

CRC cell lines are commonly used to evaluate the efficacy of immunotherapy or chemioimmunotherapy (*Nat Mater.* 2022;21(6):710-720. PMID:35606429; *Nat Commun.* 2019;10(1):1899. PMID: 31015397). In the revised manuscript, we have tested more different cancer cell lines, including human cancer cells such as HCT116 (colon cancer cells), Hela (cervical cancer cells), and MDA-MB-231 (breast cancer cells), and demonstrated that LID+US exhibited more potent tumor cell killing than conventional chemotherapy and may have broad applications (Fig.4e-f, Supplementary Figure 7).

Line 191 - supply citation of histones in nuclei being able to protect nDNA, if not, it is purely speculation, and should be clearly stated as such.

Previous studies have reported histones may protect nDNA when tumor cells are exposed to ionizing radiation, which can also generate ROS (*Cell Mol Immunol.* 2021;18(9):2211-2223. PMID: 32398808). Our data indicate that cells exposed to low-dose ultrasound-responsive chemotherapeutics can undergo a similar change. We have added the reference and rewritten the text in the revised manuscript.

Line 224 - it would be better if a natural non-forcibly expressed protein was used as epitope to prove this point. Data would be more translationally relevant.

OVA is widely used as an epitope in previous studies to monitor immune responses (*Nat Commun.* 2019;10(1):5108. PMID:31704921). We have also used tumor cells not expressing OVA and demonstrated that CD8+ T cell responses were significantly enhanced by LID+US on these models (Fig.6k, Fig.7g, i).

Line 293 - without toxicity/safety data, this dataset and the manuscript therefore is not acceptable. Note that for a drug like Dox, which is highly effective in killing tumor cells, it is the toxicity to normal tissues that forces the chief oncologist to reduce or cease further treatment. Furthermore, I would like to emphasise that in an 'acute' (for want of a better term) model like this, where animals were dosed for 40 days, the accumulation of drug is not allowed to occur to the point where it becomes toxic, even if treatment is ceased. Dox is not only toxic during dosing, but months or years

after due to its ability to accumulate in the body. It is not enough to show efficacy data alone, as lots of promising anticancer technologies stumble later on due to the lack of toxicity/safety data at the outset. Thus, it is imperative that toxicity data is shown.

We have performed safety studies and demonstrated that LID+US was safe and well tolerated (Supplementary Figure 12). It should be noted that 2 cycles of LID+US treatment are sufficient to initiate the antitumor immunity, which can efficiently prevent metastasis and relapse. Long-term use of LID+US is not required. It is the established antitumor immunity that can provide long-term protection for animals and prevent metastasis and relapse. Similar dosing regimen has been used in previous studies involving chemotherapy and photodynamic therapy (*Nat Commun.* 2016;7:12499. PMID: 27530650).

Line 293 - further to the above, it would have helped greatly if normal cells were also exposed to the technology described. Cardiomyocytes such as differentiated H9c2 would have been a good model. Note that this would do away with the need for in vivo safety evaluation, but would complement it.

We have performed in vivo studies to demonstrate that LID+US was safe and well tolerated (Supplementary Figure 12). Since the heart was not exposed to US in vivo, in vivo studies are better able to reflect the safety profile of LID+US.

Line 481 - the sc model chosen here is too simplistic and data therefore hardly clinically relevant. For instance, a sc model does not incorporate any element of metastasis, but is one solid tumour, at the wrong spot (ie. away from the colorectal area), when compared to a more clinically-relevant orthotopic model which would incorporate cells being shed from the site of injection to neighboring tissues such as liver and stomach. To put it a different way, a primary tumor like the one established in this study would be easy to resect if presented clinically, so Dox treatment would not be critical or even required. As such, there is a flaw in the design of this study, and at the most critical point - when the team is trying to prove that their technology works. Yes, it does work, rather well, in this oversimplified model. But, is it clinically-facing or -relevant? I doubt that.

We agree that the subcutaneous model is a simplified model. However, the sc model can be useful to validate the key features of formulations as reported before (*Nat Commun.* 2016;7:12499. PMID: 27530650; *Nat Mater.* 2022;21(6):710-720. PMID:35606429).

To pressure test our formulation, now we have added clinically relevant orthotopic 4T1 models, which have been widely used in previous studies (*Nat Commun.* 2019;10(1):2025. PMID:31048681).

It should be noted that the value of using LID+US was not only to directly kill primary tumors, but also to induce potent STING activation and antitumor immunity that can kill residual tumor cells (not directly killed by LID+US) and metastasized tumors (not exposed to ultrasound) and prevent relapse. These effects can not be achieved by surgically removing the primary tumor (*Sci Transl Med.* 2018;10(433):eaar1916. PMID:29563317).

Reviewer #2 - DNA damage, STING (Remarks to the Author):

This manuscript describes an interesting approach to targeting liposomal doxorubicin/indocyanine green to tumors using ultrasound. They demonstrate enhanced effect over liposomal doxorubicin and demonstrate immune activation and synergy with an anti-PD1 treatment in a murine model. The approach may have some therapeutic advantages in allowing the specific targeting of tumors.

We thank the reviewer for the positive comments.

There are, however, some issues that need addressed.

Issues:

Fig 3a/b: These would benefit from a higher magnification image. How was intracellular doxorubicin imaged? Was an antibody used?

We have used a higher magnification. Doxorubicin was fluorescent and was imaged directly without using an antibody.

Figs 3c-h: An appropriate control is missing. What effect did LID have without ultrasound? This is important as the authors noted that the acidic pH in endosomes could trigger DOX release from LID without ultrasound treatment.

We have added LID alone as a control. The activity of LID was similar to that of LD (Fig.4b-c). This is not surprising as ICG was inert in the absence of ultrasound.

Fig 4. The authors have only focused on oxidised DNA damage as the main mechanism of action. Doxorubicin is a potent topoisomerase II inhibitor and induces double stranded DNA breaks. They need to look at markers of DNA breaks in the nucleus such as H2AX foci. This will likely demonstrate nuclear DNA damage. This is important as nuclear DNA damage leads to the accumulation of cytosolic DNA in tumors thereby triggering the cGAS-STING pathway. (Harding et al Nature. 2017 Aug 24; 548(7668): 466–470). This is independent on APCs.

We have performed additional experiments to demonstrate the nuclear DNA damage after treatment LID+US. LID+US caused more nuclear DNA damage compared with LD or LI+US (Supplementary Figure 6). We have demonstrated that removing BMDCs almost abrogated IFN β secretion, indicating IFN β was mainly BMDCs rather than from tumor cells (Supplementary Figure 10). Furthermore, our data indicate that mitochondrial DNA plays a critical role in activating the STING pathway in APCs. A similar phenomenon has been reported for tumor cells exposed to ionizing radiation, which can also generate ROS (*Cell Mol Immunol.* 2021;18(9):2211-2223. PMID: 32398808). However, it should be noted our data do not exclude the possibility that other DAMPs from tumor cells or other PRRs in APCs are involved in the activation of APCs. In fact, the presence of residual type I interferon after mitochondrial DNA depletion implied that other DAMPs from dying tumor cells were also involved in the activation process (Fig. 5d).

Fig 4e: There needs to be more detail on how mitochondria were depleted in the manuscript other than a mention of ddC in the figure legend. These data do not necessarily support oxidised mitochondrial DNA as the main activator of cGAS-STING. It is equally possible that loss of mitochondrial function has an indirect

negative effect on STING/interferon signaling. Does treatment with exogenous cGAMP still activate the STING pathway in the absence of mitochondria? Do DNA damaging agents that target nuclear DNA (such as etoposide) still activate cGAS-STING?

Mitochondrial DNA in tumor cells was depleted by using 150 uM daysideoxycytidine (ddc) to treat tumor cells for 6 days. Under this condition, mtDNA was largely missing (Supplementary Figure 9). We have also shown depleting mtDNA of tumor cells can compromise IFN β secretion from BMDCs, indicating mtDNA plays a critical role in activating the STING pathway in BMDCs (Fig.5d).

In our system, STING activation mainly occurs in BMDCs rather than in tumor cells (Supplementary Figure 10). The BMDCs we use can be directly activated by using exogenous cGAMP in the absence of tumor cells (data not shown). Since we did not deplete mitochondrial DNA of BMDCs, the mitochondrial function of BMDCs was normal and should not affect the STING signaling in BMDCs. While we used the topoisomerase inhibitor DOX as a model drug in this study, other topoisomerase inhibitors such as etoposide can have similar effects (*Cell Res.* 2020;30(8):639-648. PMID: 32541866). We have mentioned this in the discussion.

Fig 5: The authors describe the RAW-lucia ISG cell line as a reporter for cGAS/STING activation. This is inaccurate. It is a reporter for interferon alpha and beta release following potential activation of several pattern recognition receptors. It is not specific to cGAS/STING activation.

We have updated the description. We have also added RNA seq data for WT BMDCs and STING KO BMDCs to confirm the STING-dependent activation of BMDCs.

Fig 5i. It is unclear where STING activation is occurring in response to the treatment. Is it in the cancer cells or dendritic cells or both? As mentioned above there is good evidence that DNA damaging agents activate cGAS-STING directly in tumors resulting in cytokine release and T cell recruitment (this is independent of APCs). The authors therefore need to look at cGAS-STING (IRF3 and TBK1 activation) following LID/US treatment in MC38 cells in-vitro. STING should then be knocked out of this cell line to show dependence.

We have added new data to show that tumor cells alone fail to secrete IFN-beta (a sign of STING activation) in the absence of BMDC (Supplementary Figure 10). These results clearly indicate STING activation is mainly occurring in dendritic cells rather than in tumor cells. Furthermore, we have also shown that LID+US did not work well on STING KO mice bearing WT MC38 tumors, but worked well on WT C57BL/6 mice bearing STING KO MC38 tumor cells (Fig.6l-m), further indicating STING signaling in tumor cells was not critical for animals treated with LID+US.

Fig 6: The authors need to explain if in the bilateral tumor experiments both tumors were treated with US. It would be more clinically relevant if metastatic sites regressed following treatment of the primary tumor only, as often metastatic sites are either inaccessible or too numerous to treat directly.

Only the primary tumor was treated with US. The distal tumor was not treated with US. We have clarified this in the revised manuscript.

Reviewer #3 - nanoparticles, activatable liposomes (Remarks to the Author):

This contribution describes a study of liposomes containing both indocyanine green as a 'sonosensitizer' and doxorubicin as an anti-tumor agent and their ability to inhibit tumor cell growth in culture and in xenograft models when exposed to ultrasound. Although there are extensive experiments performed showing modest tumor suppression cell culture and more significant tumor remissions in vivo, there are two major flaws in this reviewer's view that make this work unpublishable in its current form.

1. The mode of action of sonosensitization is poorly understood in general and this contribution does nothing to improve our understanding of the physical basis of ROS generation. Although there are data presented showing modest biological impacts as a function of sonoexcitation time, there are no data presented showing how the DOPE-ICG loading impacts the function of the liposome formulations on cells. Since the in vitro experiments are conducted with 24 hour exposures of the cells to the liposome formulations, there is ample time for the agents to be internalized by the cells and repurposed to other organelles prior to sonoexcitation. It is not clear that the ROS production is a result of the intact formulation or cellular redistribution of the liposomal components. Higher resolution and time-dependent studies are needed to better understand this system.

We appreciate the reviewer's helpful comments. We have performed high resolution and time-dependent studies to confirm that DOPE-ICG was mostly located in endosomes/lysosomes prior to sonoexcitation (Supplementary Figure 4). The distinct distribution profile of DOPE-ICG and ROS indicates that ROS can escape from endosomes and reach other parts of cells.

2. There is an unacceptable lack of an important control throughout the study -- that is, the impact of liposomal doxorubicin + US in the absence of ICG. It is conceivable that the nanocrystalline DOX is sonochemically active by itself and is responsible for the reported effects by altering its dissolution profile once in the cell and/or tumor tissue environment in vivo. The LI + US control was reported, however, a LD + US control must be included in the study to more clearly demonstrate the necessity of including ICG in the formulation to achieve the reported bioactivity.

We have shown in Fig 3a that LI+US and LID+US were similar to each other in terms of ROS generation, indicating the presence of DOX did not interfere with ROS generation and DOX was not sonochemically active, at least under the dose we used. Moreover, we have added LD+US both in vitro and in vivo and demonstrated that LD+US was much less effective than LID+US (Fig.3b,c,e,f, Supplementary Figure 7, Supplementary Figure 11), indicating ICG was required in the formation to achieve the reported therapeutic efficacy.

Reviewers' Comments:

Reviewer #2:

Remarks to the Author:

The authors have provided additional data and clarifications that have enhanced the manuscript. The data demonstrating metastatic sites respond to this treatment, independent of direct ultrasound, is an important clarification and increases the potential clinical applicability of this therapeutic approach.

I have no further comments.

Reviewer #4:

Remarks to the Author:

In the manuscript "Ultrasound responsive low-dose chemotherapy uncovers the importance of oxidized tumor mitochondrial DNA in activating antitumor immunity" Wang et al. show that liposomes loaded with the lipid-modified sonosensitizer indocyanine green and low-dose doxorubicin (LID) generate reactive oxygen species in response to ultrasound (US). This dual treatment increased the nuclear delivery of doxorubicin within tumor cells and induced mitochondrial tumor DNA oxidation. Their experiments suggest that oxidized mitochondrial tumor DNA contributes to the activation of STING in antigen presenting cells (APCs) and APC-mediated T cell stimulation leading to more efficient anti-tumor responses. In support of their claims, depletion of mitochondrial tumor DNA or deletion of STING in APCs significantly compromised the activation of APCs. Interestingly, inhibition of PD-L1 further enhanced efficacy and led to a regression of bilateral tumors in three inflamed murine tumor models (MC38, CT26, and 4T1 tumors). The findings and claims of the authors are generally well supported by the data in the manuscript and the data are presented in a logical and clear manner. Addressing the following concerns might further increase the appeal of the manuscript in my opinion:

Major concerns

- Lack of in vivo proof that increased antigen cross-presentation by APCs contributes to the improved anti-tumor response upon LID + US treatment. As the authors already use the OVA system a transfer of naïve OT-I T cells might help to at least partially address this question?
- Evidence that transfer of oxidized mitochondrial tumor DNA to APCs is mediating the effects and not tumoral cGAMP. This might be addressed by using cGAS deficient tumor cells in the experiments? In general, I am a little surprised that the effects appear to mainly depend on oxidized mitochondrial DNA and not damaged genomic DNA, which is also generated upon doxorubin treatment? Could the authors comment or investigate?
- The authors used inflamed tumor models. It would be important to show the effects in less inflamed tumor models such as B16.

Minor concerns

- Liposomes tend to accumulate in liver and kidney. In the experiments shown in the manuscript the authors don't observe such accumulation. Maybe it would be worth discussing the reason for this? I suppose the time point of the read-out was chosen accordingly?
- Figure 1: I wonder if this figure could go towards the end of the manuscript?
- Figure 4: Could all the data be shown in the figures rather than representative experiments?
- Figure 5d: As IFN β ELISAs are not very sensitive, the authors might wish to also test phosphorylation of TBK1 and IRF3?
- Figure 5f. Could all the experimental data be summarized in the figure? Maybe the data could be presented as fold increase of MFI over untreated? It would also be important to determine the functional relevance of the increase of SIINFEKL presentation by analyzing activation of OT-I T cells.
- Figure 6i: ISG levels should also be measured to better understand the strength of the IFN response.
- Supplementary figure 15 is not referenced in the manuscript?

Reviewer #5:

Remarks to the Author:

In this study, the authors constructed a lipid modified liposome system that co-delivers sonosensitizer indocyanine green and low-dose doxorubicin for effective strategies of cancer immunotherapy. The authors have conducted substantial experiments and demonstrated the significant efficacy *in vivo*. Still, there are specific issues remaining to be addressed.

(1) For formulation characterization, drug release study in Supplementary Figure 5 is insufficient. The time-dependent release curves for both ICG and DOX should be supplemented under different conditions over time. Furthermore, the detailed description for fabrication of LID with a fixed drug ration (8:1, 4:1, 2:1, 1:1) should be provided. Is this the exact drug ratio within the liposome system? Also, the stabilities of the delivery system need to be studied.

(2) The authors claimed that ROS generation by LI+US facilitated endosomal escape of DOX, which resulted in the promoted nuclear distribution of DOX. Please provide direct evidence of endosome rupture or increased permeability of DOX across the endosome after LI+US treatment. Also, the increased localization of DOX in nuclei could be due to the accelerated DOX release or leakage upon the LI+US. Overall, the mechanism for intracellular fate of drug and signal transduction is still unclear.

(3) The greatest concern for the rationale is that how DOX contributes to the activation of immune system. It seems that the authors only ascribed the immune activation to "the importance of oxidized tumor mitochondrial DNA", which is mainly generated by LI+US (Figure 5a). However, DOX itself is also a potent inducer of immunogenic cell death (Science advances, 2018, 4(4): eaao1736), which triggers calreticulin (CRT) exposed on cell surface, ATP and HMGB1 release that promotes activation of DCs and trigger antigen-specific T cell responses. Thus, the ICD-inducing effect of DOX could not be ignored. Also, could LI+US promote the ICD-inducing capacity of DOX? It is not clear that the superior antitumor effects of LID+US is a result of elevated ICD induced by Dox action on ROS-sensitive tumor cells or effective transport of oxidized mitochondrial DNA to APCs as described by the authors in the manuscript. Specific experiments need to be designed.

(4) In figure 6j,7f, and 7h, the contour plot showing CD8 (x-axis) and CD4 (y-axis) has lots of CD3+ T cells that are not CD8 or CD4 T cells. What are those CD3+T cells if they are not either CD8+T cells or CD4+T cells? This applies to the contour plot in Supplementary Figure 14. In addition, the promising anti-tumor effect *in vivo* solely depends on the immune activation or also includes the cytotoxicity of ICG and DOX? LID treatment combined with CD8+ T cell depletion should be performed as the control.

Reviewer #6:

Remarks to the Author:

This work describes an antitumor study of liposomes containing indocyanine green and conventional chemotherapy DOX for improved therapy efficiency. Actually, similar construction of synergistic nanodrug has been widely reported in past decades. To my mind, the main idea this work wants to highlight is that ROS generated by sonosensitizer indocyanine green could amplify the cytotoxicity of DOX by promoting DOX delivery to nuclei, and simultaneously oxidizing tumor mitochondrial DNA for antitumor immunity. Although this work provided some data showing modest tumor suppression cell culture and significant tumor remissions *in vivo*, there are several flaws that need to be sufficiently addressed before finally being published.

1. It is conceivable that the synergistic effect of ROS and chemotherapy can certainly improve therapeutic efficiency. However, the most attractive effect of synergistic therapy of ROS and DOX is significantly decreasing the dose of DOX. The therapeutic efficiency in the condition of lower-dose DOX should be discussed, and the detailed concentration range of Dox should also be discussed which would get a similar anticancer effect to the dose used in the current stage.

2. The indicated DOX nanocrystals illustrated in Fig.2d are quite arbitrary. More characterization should be provided, such as high-resolution STEM, and the formation mechanism should be demonstrated.

3. It is incredible that there is so big difference in DOX delivery to nuclei with and without US treatment after 24 h incubation. Is it a US-responsive release of DOX? If yes, all the control experiments in this work should be redesigned.

We highly appreciate the reviewers' thorough review of our manuscript and their constructive comments. We have carefully addressed the reviewers' concerns in the following point-by-point responses and revised the manuscript accordingly.

Reviewers' comments:

Reviewer #2 (Remarks to the Author):

The authors have provided additional data and clarifications that have enhanced the manuscript. The data demonstrating metastatic sites respond to this treatment, independent of direct ultrasound, is an important clarification and increases the potential clinical applicability of this therapeutic approach.

I have no further comments.

We thank the reviewer for the positive endorsement of our work.

Reviewer #4 - APCs (Remarks to the Author):

In the manuscript "Ultrasound responsive low-dose chemotherapy uncovers the importance of oxidized tumor mitochondrial DNA in activating antitumor immunity" Wang et al. show that liposomes loaded with the lipid-modified sonosensitizer indocyanine green and low-dose doxorubicin (LID) generate reactive oxygen species in response to ultrasound (US). This dual treatment increased the nuclear delivery of doxorubicin within tumor cells and induced mitochondrial tumor DNA oxidation. Their experiments suggest that oxidized mitochondrial tumor DNA contributes to the activation of STING in antigen presenting cells (APCs) and APC-mediated T cell stimulation leading to more efficient anti-tumor responses. In support of their claims, depletion of mitochondrial tumor DNA or deletion of STING in APCs significantly compromised the activation of APCs. Interestingly, inhibition of PD-L1 further enhanced efficacy and led to a regression of bilateral tumors in three inflamed murine tumor models (MC38, CT26, and 4T1 tumors). The findings and claims of the authors are generally well supported by the data in the manuscript and the data are presented in a logical and clear manner. Addressing the following concerns might further increase the appeal of the manuscript in my opinion:

We appreciate the reviewer's helpful comments.

Major concerns

- Lack of in vivo proof that increased antigen cross-presentation by APCs contributes to the improved anti-tumor response upon LID + US treatment. As the authors already use the OVA system a transfer of naïve OT-I T cells might help to at least partially address this question?

In the MC38-OVA tumor model, OVA+ tumor cells undergoing immunogenic cell death (ICD) can activate APCs and induce the expansion of SIINFEKL-specific CD8+ T cells (bearing the same TCR of OT-I T cells) (Nat Commun. 2019;10(1):5108. PMID: 31704921). We harvested the tumor tissue and used peptide-MHC tetramer tagged with PE (H-2K^b-restricted SIINFEKL) to analyze the percent of SIINFEKL-specific CD8+ T cells. LD and LI+US exhibited similar levels of SIINFEKL-specific CD8+ T cells compared with the untreated animals, while LID+US exhibited ~30% SIINFEKL-specific CD8+ T cells (Fig.5I). These results indicate that the increased antigen cross-presentation by APCs upon LID+US treatment is translated to more efficient expansion of SIINFEKL-specific CD8+ T cells.

- Evidence that transfer of oxidized mitochondrial tumor DNA to APCs is mediating the effects and not tumoral cGAMP. This might be addressed by using cGAS deficient tumor cells in the experiments? In general, I am a little surprised that the effects appear to mainly depend on oxidized mitochondrial DNA and not damaged genomic DNA, which is also generated upon doxorubicin treatment? Could the authors comment or investigate?

We have performed experiments to confirm whether the STING activation in BMDC was mediated by tumor DNA or by tumoral cGAMP. For the LID+US group, knocking out cGAS from MC38 tumor cells

did not compromise the IFN β secretion from BMDC, indicating cGAS in MC38 was not required to mediate the APC activation. In contrast, knocking out cGAS from BMDC abrogated IFN β secretion from BMDC (Supplementary Figure 13b). As tumoral cGAMP can directly induce IFN β secretion from BMDC without requiring functional cGAS in BMDC, these results indicate that tumor DNA, rather than tumoral cGAMP, mainly mediated the effect. Previous studies have reported that histones may protect genomic DNA when tumor cells are exposed to ROS generated by ionizing radiation (Cell Mol Immunol. 2021;18(9):2211-2223. PMID: 32398808). It is possible that the presence of additional barriers prevented the oxidation of genomic DNA and transfer to APCs. While our data suggest that oxidized tumor mitochondrial DNA is important in STING-mediated antitumor immunity after treatment by LID+US, we can not rule out that other DAMPs (e.g., HMGB1 and genomic DNA) may also play a role. The effect of different DAMPs is highly dependent on the way how tumor cells are treated.

- The authors used inflamed tumor models. It would be important to show the effects in less inflamed tumor models such as B16.

We have tested the therapeutic effect in bilateral B16F10 tumors. Compared with LID+US and α PD-L1, LID+US+ α PD-L1 exhibited a more potent therapeutic effect on primary tumors (exposed to ultrasound) and distant tumors (not exposed to ultrasound) (Supplementary Figure 25).

Minor concerns

- Liposomes tend to accumulate in liver and kidney. In the experiments shown in the manuscript the authors don't observe such accumulation. Maybe it would be worth discussing the reason for this? I suppose the time point of the read-out was chosen accordingly?

In the manuscript, we used the IVIS optical imaging system to perform whole-body imaging in order to continuously monitor the accumulation of liposomes in the tumor over time. The goal is to identify a time range during which the sonosensitizer accumulation in the tumor reached a plateau such that ultrasound can activate the sonosensitizer at the maximal level. Due to the limited penetration depth of the laser of IVIS, fluorescence signals in deep organs such as the liver and kidney can not be detected efficiently (Nat Nanotechnol. 2021;16(11):1260-1270. PMID: 34594005). Indeed, liposomes can accumulate in the liver, spleen, and other normal organs (Supplementary Figure 15). The goal of this study is not to directly change the distribution profile of liposomes, but to activate LID in the tumor to induce targeted cytotoxicity and activation of antitumor immunity. LID in normal organs was not activated due to the absence of ultrasound and no toxicity was observed in these normal organs (Supplementary Figure 17).

- Figure 1: I wonder if this figure could go towards the end of the manuscript?

We have moved Figure 1 to the end of the manuscript (Figure 8).

- Figure 4: Could all the data be shown in the figures rather than representative experiments?

We have shown all the data (see Figure 3).

- Figure 5d: As IFN β ELISAs are not very sensitive, the authors might wish to also test phosphorylation of TBK1 and IRF3?

We have tested the phosphorylation of TBK1 and IRF3. The data are shown in Supplementary Figure 12.

- Figure 5f. Could all the experimental data be summarized in the figure? Maybe the data could be presented as fold increase of MFI over untreated? It would also be important to determine the functional relevance of the increase of SIINFEKL presentation by analyzing activation of OT-I T cells. We have shown all the data (Figure 4f-g). We have also analyzed the activation of OT-I T cells (Fig.5i), which was positively correlated with the antigen cross-presentation.

- Figure 6i: ISG levels should also be measured to better understand the strength of the IFN response. We have measured the ISG levels. As LD and LI+US failed to significantly induce STING activation in the tumor, we focused on LID+US in terms of ISG levels. Compared with the untreated group, LID+US significantly upregulated ISG levels. (Supplementary Figure 19).

- Supplementary figure 15 is not referenced in the manuscript?

We have now referenced this figure in the manuscript.

Reviewer #5 - Replacement for R#1 - DOX delivery (Remarks to the Author):

In this study, the authors constructed a lipid modified liposome system that co-delivers sonosensitizer indocyanine green and low-dose doxorubicin for effective strategies of cancer immunotherapy. The authors have conducted substantial experiments and demonstrated the significant efficacy in vivo. Still, there are specific issues remaining to be addressed.

We appreciate the reviewer's helpful comments.

(1) For formulation characterization, drug release study in Supplementary Figure 5 is insufficient. The time-dependent release curves for both ICG and DOX should be supplemented under different conditions over time. Furthermore, the detailed description for fabrication of LID with a fixed drug ratio (8:1, 4:1, 2:1, 1:1) should be provided. Is this the exact drug ratio within the liposome system? Also, the stabilities of the delivery system need to be studied.

We have added time-dependent release curves. Under neutral pH, LID released minimal DOX in the absence or presence of US. Under acidic pH (endosomal pH), LID efficiently released DOX and US only slightly boosted DOX release (Supplementary Figure 6a). As ICG was covalently attached to the phospholipid, LID released minimal ICG under neutral pH or acidic pH. US boosted ICG release at acidic pH much more than that of neutral pH, but the overall release was much lower than DOX (Supplementary Figure 6b). We have added a more detailed description for the fabrication of LID in the revised manuscript. The ratios represent the exact drug ratio within the liposome system. LID was stable in the presence of serum, with minimal release of DOX and ICG at 37 °C over 48 h (Supplementary Figure 3).

(2) The authors claimed that ROS generation by LI+US facilitated endosomal escape of DOX, which resulted in the promoted nuclear distribution of DOX. Please provide direct evidence of endosome rupture or increased permeability of DOX across the endosome after LI+US treatment. Also, the increased localization of DOX in nuclei could be due to the accelerated DOX release or leakage upon the LI+US. Overall, the mechanism for intracellular fate of drug and signal transduction is still unclear.

We have performed additional experiments to uncover how LI+US promoted the nuclear distribution of DOX. Under neutral pH, LID released minimal DOX in the absence or presence of ultrasound. Under acidic pH (endosomal pH), LID efficiently released DOX and US only slightly boosted DOX release (Supplementary Figure 6a). These results imply that the accelerated DOX release triggered by LI+US was not the major reason for the nuclear distribution of DOX. In fact, as an organic base, released DOX was protonated within acidic endosomes and unable to efficiently cross the endosomal membrane (J Control Release. 2018; 288: 96-110. PMID:30184465). To further analyze the permeability of the endosomal membrane, we took advantage of a fluorescent dye Acridine Orange (AO), which was able to cross the plasma membrane of live cells and exhibit red fluorescence within acidic endosomes but green fluorescence within neutral nuclei. Importantly, AO was protonated within acidic endosomes and unable to cross the endosomal membrane unless the permeability of endosomes was increased, so the reduction of red AO fluorescence has been used to evaluate the

endosomal membrane permeability (Cell Death Dis. 2021;12(1):80. PMID: 33441536). We treated MC38 cells with LI or LI+US and then added AO to MC38 cells. We found AO+LI led to bright red fluorescence in endosomes, while AO+LI+US did not exhibit red fluorescence in endosomes (Supplementary Figure 7), indicating LI+US enhanced the permeability of the endosomal membrane.

(3) The greatest concern for the rationale is that how DOX contributes to the activation of immune system. It seems that the authors only ascribed the immune activation to “the importance of oxidized tumor mitochondrial DNA”, which is mainly generated by LI+US (Figure 5a). However, DOX itself is also a potent inducer of immunogenic cell death (Science advances, 2018, 4(4): eaao1736), which triggers calreticulin (CRT) exposed on cell surface, ATP and HMGB1 release that promotes activation of DCs and trigger antigen-specific T cell responses. Thus, the ICD-inducing effect of DOX could not be ignored. Also, could LI+US promote the ICD-inducing capacity of DOX? It is not clear that the superior antitumor effects of LID+US is a result of elevated ICD induced by Dox action on ROS-sensitive tumor cells or effective transport of oxidized mitochondrial DNA to APCs as described by the authors in the manuscript. Specific experiments need to be designed.

We have clarified the mechanism of action for LID+US. The key features of chemotherapeutics-induced immunogenic cell death (ICD) include CRT exposure on dying tumor cells and release of damage-associated molecular patterns (DAMPs) such as HMGB1, ATP, and the recently identified DAMP tumor DNA (Cell Res. 2020;30(8):639-648. PMID: 32541866). Among all the DAMPs, DNA derived from tumor cells undergoing ICD is an important one as it can activate the cGAS-STING pathway to promote T cell immunity. However, conventional liposomal DOX (denoted as LD, similar to the FDA approved Doxil[®]) can be sequestered in endosomes/lysosomes and fail to efficiently kill tumor cells, resulting in poor transfer of DAMPs (especially DNA) to APCs. Furthermore, DNA can be degraded in APCs. We demonstrated that LID+US-induced ROS promoted the endosomal escape of DOX by enhancing the endosomal membrane permeability, resulting in efficient nuclear delivery of DOX within tumor cells, while traditional liposomal DOX (LD) was sequestered in endosomes. Importantly, LID+US-induced ROS also caused mitochondrial DNA oxidation that can better resist degradation. Consequently, LID+US efficiently killed tumor cells, increased tumor mitochondrial DNA transfer to APCs for STING activation. Removing any component from LID+US can compromise tumor cell killing and/or DNA oxidation/transfer, thus compromising the APC activation. Moreover, we have performed additional experiments to demonstrate that LID+US upregulated other ICD markers such as CRT and HMGB1, with levels similar (for CRT) or higher (for HMGB1) compared with LD and LI+US (Supplementary Figure 14), so multiple DAMPs may have contributed to the activation of APCs. Depleting tumor mitochondrial DNA or knocking out cGAS or STING in APCs dramatically compromised the activation of APCs, indicating that tumor mitochondrial DNA is indeed an important DAMP. While our data suggest that oxidized tumor mitochondrial DNA is important in STING-mediated antitumor immunity after treatment by LID+US, we can not rule out that other DAMPs (e.g., HMGB1 and genomic DNA) may also play a role. The effect of different DAMPs is highly dependent on the way how tumor cells are treated.

(4) In figure 6j, 7f, and 7h, the contour plot showing CD8 (x-axis) and CD4 (y-axis) has lots of CD3+ T cells that are not CD8 or CD4 T cells. What are those CD3+T cells if they are not either CD8+T cells or CD4+T cells? This applies to the contour plot in Supplementary Figure 14. In addition, the promising anti-tumor effect in vivo solely depends on the immune activation or also includes the cytotoxicity of ICG and DOX? LID treatment combined with CD8+ T cell depletion should be performed as the control. CD3+CD8-CD4- cells have been observed in other studies as well (Nat Nanotechnol. 2019;14(3):269-

278. PMID: 30664751). They are known as CD3+CD8-CD4- double negative T cells. While the exact function of these cells is beyond the scope of this manuscript, previous studies have shown that these cells can have antitumor or protumor effects, depending on the context (Front Immunol. 2022;13:816005. PMID: 35222392). We have performed additional experiments to test LID+US combined with CD8+ T cell depletion. Depleting CD8+ T cells significantly compromised the therapeutic effect of LID+US (Supplementary Figure 22). Together with the compromised therapeutic effect of LID+US on STING KO mice (Fig.5m), these data indicate that the host immune activation is important to mediate the antitumor effect in vivo.

Reviewer #6 - Replacement for R#3 - US activatable drug delivery (Remarks to the Author):

This work describes an antitumor study of liposomes containing indocyanine green and conventional chemotherapy DOX for improved therapy efficiency. Actually, similar construction of synergistic nanodrug has been widely reported in past decades. To my mind, the main idea this work wants to highlight is that ROS generated by sonosensitizer indocyanine green could amplify the cytotoxicity of DOX by promoting DOX delivery to nuclei, and simultaneously oxidizing tumor mitochondrial DNA for antitumor immunity. Although this work provided some data showing modest tumor suppression cell culture and significant tumor remissions in vivo, there are several flaws that need to be sufficiently addressed before finally being published.

We appreciate the reviewer's helpful comments.

1. It is conceivable that the synergistic effect of ROS and chemotherapy can certainly improve therapeutic efficiency. However, the most attractive effect of synergistic therapy of ROS and DOX is significantly decreasing the dose of DOX. The therapeutic efficiency in the condition of lower-dose DOX should be discussed, and the detailed concentration range of Dox should also be discussed which would get a similar anticancer effect to the dose used in the current stage.

We have added a new discussion in the revised manuscript. Briefly, chemotherapy alone typically requires a high dose of DOX 5 mg/kg (Adv Sci. 2022;10(1):e2205247. PMID:36453573) to induce a meaningful therapeutic effect. In our study, the liposomal DOX (denoted as LD, similar to commercially available Doxil[®]) dose was 0.5 mg/kg, which as chemotherapy alone failed to induce a meaningful therapeutic effect. In contrast, LID+US administered at 0.5 mg/kg of DOX induced a potent therapeutic effect. We demonstrated, for the first time, that this was partially achieved by promoting the nuclear delivery of DOX to tumor cells, inducing tumor mitochondrial DNA oxidation, and facilitating the transfer of oxidated tumor DNA to APCs to maximize cGAS-STING activation in APCs, which ultimately induced strong antitumor immunity and potent therapeutic effects.

2. The indicated DOX nanocrystals illustrated in Fig.2d are quite arbitrary. More characterization should be provided, such as high-resolution STEM, and the formation mechanism should be demonstrated.

We loaded DOX into liposomes using the "active loading approach" following previous reports (J Control Release. 2018; 288: 96-110. PMID:30184465). Briefly, blank liposomes were prepared with 250 mM (NH₄)₂SO₄, followed by the removal of external (NH₄)₂SO₄ using size exclusion chromatography to establish the transmembrane gradient. DOX was incubated with the blank liposomes at 55 °C (above the transition temperature of DPPC). When NH₃ escaped liposomes, one H⁺ was produced and retained in the liposome, resulting in an acidic core. When DOX diffused into the liposome, it became protonated and trapped within the liposome. As DOX loading into the liposome transiently increased the internal pH, it further increased the level of ammonia and created

more H⁺, allowing more DOX to be loaded into the liposome. Ultimately, DOX forms a crystalline precipitate due to the presence of sulfate anions inside the liposome. The detailed formation mechanism is now shown in Fig. 1. We have used the state-of-art cryo ET to characterize LID, which exhibited DOX crystalline precipitate (nanocrystal) in the core (Supplementary Movie 1).

3. It is incredible that there is so big difference in DOX delivery to nuclei with and without US treatment after 24 h incubation. Is it a US-responsive release of DOX? If yes, all the control experiments in this work should be redesigned.

We have performed additional experiments to uncover how LI+US promoted the nuclear distribution of DOX. Under neutral pH, LID released minimal DOX in the absence or presence of ultrasound. Under acidic pH (endosomal pH), LID efficiently released DOX and US only slightly boosted DOX release (Supplementary Figure 6a). These results imply that the accelerated DOX release triggered by LI+US was not the major reason for the nuclear distribution of DOX. In fact, as an organic base, released DOX was protonated within acidic endosomes and unable to efficiently cross the endosomal membrane (J Control Release. 2018; 288: 96-110. PMID:30184465). To further analyze the permeability of the endosomal membrane, we took advantage of a fluorescent dye Acridine Orange (AO), which was able to cross the plasma membrane of live cells and exhibit red fluorescence within acidic endosomes but green fluorescence within neutral nuclei. Importantly, AO was protonated within acidic endosomes and unable to cross the endosomal membrane unless the permeability of endosomes was increased, so the reduction of red AO fluorescence has been used to evaluate the endosomal membrane permeability (Cell Death Dis. 2021;12(1):80. PMID: 33441536). We treated MC38 cells with LI or LI+US and then added AO to MC38 cells. We found AO+LI led to bright red fluorescence in endosomes, while AO+LI+US did not exhibit red fluorescence in endosomes (Supplementary Figure 7), indicating LI+US enhanced the permeability of the endosomal membrane.

Reviewers' Comments:

Reviewer #4:

Remarks to the Author:

The authors have adequately addressed my concerns in the revised version.

Reviewer #5:

Remarks to the Author:

The authors have provided additional data and clarifications to improve the mechanism understanding of the drug delivery system as well as the quality of the manuscript.

Reviewer #6:

Remarks to the Author:

The additional data and explanations that were provided have fully addressed all the questions and concerns I had regarding the work. I have no further comments.

REVIEWERS' COMMENTS

Reviewer #4 (Remarks to the Author):

The authors have adequately addressed my concerns in the revised version.

Response: We appreciate the positive comments of the reviewer.

Reviewer #5 (Remarks to the Author):

The authors have provided additional data and clarifications to improve the mechanism understanding of the drug delivery system as well as the quality of the manuscript.

Response: We appreciate the positive comments of the reviewer.

Reviewer #6 (Remarks to the Author):

The additional data and explanations that were provided have fully addressed all the questions and concerns I had regarding the work. I have no further comments.

Response: We appreciate the positive comments of the reviewer.